# VapC toxins promote the pathogenesis of *Rickettsia heilongjiangensis* by cleaving essential RNAs from both *Rickettsia* and its host

Yan Liu[1,2], Weiting Zhou[1], Jiaying Zhao[1], Qingyin Shi[1], Yu Xin[1], Xuan Ouyang[3], Yonghui Yu[3], Jun Jiao[3], Maozhang He[1], Yajun Song[3]*, Kehan Xu[1]*

**1** Anhui Province Key Laboratory of Zoonoses, Department of Microbiology, School of Basic Medical Sciences, Anhui Medical University, Hefei, China, **2** School of Public Health, Anhui University of Science and Technology, Hefei, China, **3** State Key Laboratory of Pathogen and Biosecurity, Academy of Military Medical Sciences, Beijing, China

* songyj@bmi.ac.cn (YS); kehan@ahmu.edu.cn (KX)

## Abstract

Toxin-antitoxin (TA) modules enable bacteria to persist under stressful environments. However, they are typically absent from host-associated prokaryotes due to their potential host toxicity. Here, the obligate intracellular bacterium spotted fever group (SFG) rickettsiae, which causes mild to severe human illness, was shown to harbor two *vapBC* TA modules. One of the *vapBC* modules (*vapBC*1) is crucial for *Rickettsia* to withstand accumulated host reactive oxidative species (ROS), via induction of bacterial dormancy through cleavage on the anti-codon loop of tRNA$^{fMet}$, thereby facilitating intracellular survival and infection in a mouse model. Another *vapBC* module (*vapBC*2) was found to be activated and toxin exposed to host cytoplasm, contributing to *Rickettsia*'s virulence and adaptability in its human host by non-specifically degrading host rRNAs rather than regulating rickettsial growth. Recognition of these rickettsial effectors contributes to understanding the intracellular adaptability and pathogenicity of all host-associated pathogens that harbor TA modules.

## Author summary

Vascular endothelial cells employ ROS as a common and efficient defense against pathogen infections. However, little is known about how those obligatory intracellular bacteria combat the oxidative attack that develops inside their host. Here, we demonstrated how the obligate intracellular bacterium *Rickettsia*'s *vapBC*1 TA module induces bacterial persisters in response to their host ROS threat, promoting rickettsial survival and, eventually, making infection establishment in mice easier. Notably, unspecific degradation of host ribosomal RNAs by free VapC toxin exposed to the host cytosol may alter host metabolism and promote bacterial survival. To the best of our knowledge, this is the first study to elucidate

**Data availability statement:** All data are included in the manuscript and its Supporting Information files. The raw data supporting the figures and tables are publicly available on Figshare at the following DOI: https://doi.org/10.6084/m9.figshare.29362013.

**Funding:** This work was supported by the National Natural Science Foundation of China (82002156 to K.X. and 81571963 to Y.L.) and the Natural Science Foundation of Anhui Provincial Department of Education (2024AH050743 to K.X.).The funders had no role in study design, data collection and analysis, decision to publish, or preparation of the manuscript.

**Competing interests:** The authors have declared that no competing interests exist.

TA function in an obligatory intracellular bacterium. Moreover, for all host-associated bacteria that possess *vapBC* TA modules, disrupting the impact of VapC toxins may be a viable anti-infection therapy.

## Introduction

Spotted fever group (SFG) rickettsiae are Gram-negative, obligate intracellular bacteria responsible for a series of vascular diseases that are termed "*Rickettsial* vasculitis" [1]. SFG rickettsiae that can cause disease in humans include *R. rickettsii*, the causative agent of Rocky Mountain spotted fever (RMSF), a most severe rickettsial disease characterized by high fever, organ failure, neurological symptoms, and high mortality if not treated with adequate antibiotics [2]. Another important member is *R. conorii*, responsible for Mediterranean spotted fever (MSF), a milder disease with a lower fatality than RMSF [3]. Our earlier study isolated an *R. heilongjiangensis* strain B8 (*Rh*-B8) and characterized it as a subgroup of *R. conorii*, which causes Far-Eastern spotted fever (FESF) and is widespread in northeastern and central China [4].

In humans and well-established animal infection models, SFG rickettsiae favored invading vascular endothelial cells (ECs) [5]. On the other hand, producing reactive oxidative species (ROS) is one of the most fundamental defenses mammalian cells have developed to combat invading pathogens [6–9]. Upon infection, all types of vascular cells, including vascular endothelial cells, are prone to producing ROS, including superoxide ($\cdot O_2^-$), hydrogen peroxide ($H_2O_2$), and hydroxyl radical ($\cdot OH$) [10]. These radicals are highly deleterious to invading pathogens and can directly damage DNA, proteins, and lipids [11], or indirectly damage DNA by oxidizing the nucleotide pool thereby hindering its subsequent incorporation into RNA and DNA [12]. However, it remains unclear how *Rickettsia* sustains the host ROS attack. Moreover, excessive intracellular ROS may also result in severe oxidative stress and host cell arrest, which has been linked to the ECs's injury in a variety of vascular diseases [13]. Previous research on *R. rickettsi* identified extensive host cell membrane dilatation and structural cytoskeleton disruption, which were thought to be the signs of excessive intracellular peroxides and superoxide radicals [14]. For an obligatory intracellular bacteria *Rickettsia*, it would be more critical for its interaction with host ROS production since host cells that die quickly upon infection do not aid in rickettsial growth.

Toxin-antitoxin (TA) modules are involved in a series of physiological functions, such as bacterial growth control, programmed cell death, and gene expression regulation, enabling bacteria to overcome environmental challenges [15]. Besides, TA toxins were also suggested as a secreted factor that could aid in the regulation of surrounding bacterial populations and eukaryotic cells [16–19]. A typical TA module comprises two elements, the toxin, which possesses the "toxic effect" by inhibiting an important cellular function, and an antitoxin that counteracts this effect. The regulatory circuit is initiated by the release of toxins based on the fragility of antitoxins, which is partially controlled by environmental signals. Together, this creates an

environmentally regulated stochastic equilibrium for bacteria that switches between "growing" and "nongrowing" phenotypes. Prokaryotic TA modules are classified into five types according to the antitoxin's mechanism of action, which can function as a protein (types II, IV, and V) or an RNA (types I and III). Type II TA system is the most abundant and well-characterized in bacteria and archaea, including modules *relBE*, *higBA*, *mazEF*, *ccdAB*, *vapBC*, *parDE*, *phd–doc*, *hicAB*, and *yoeB–yefM* [20]. Among these, *vapBC* operon codes for the VapC toxin, a PIN (pilT amino-terminal) domain endoribonuclease with a diverse range of intracellular RNA targets and a highly conserved active site architecture [21,22]. Because its coding gene is often located upstream of the toxin-coding gene and immediately after the promoter, VapB, its homologous antitoxin, has a higher expression level than VapC and can neutralize its toxicity under normal physiological conditions. Additionally, by directly binding to the promoter region, this antitoxin-toxin complex can subsequently block the transcription of *vapBC* operon [23].

Because of the potential toxicity of TA toxins for eukaryotic cells, TA modules were thought to be advantageous primarily to free-living microbes and incompatible with obligate intracellular prokaryotes. Genomic analysis showed that TA modules are virtually absent in most obligate intracellular organisms [24]. However, further investigation across *Rickettsia* species revealed that TA modules are rarely present in the typhus group (TG) rickettsiae yet are relatively abundant in the SFG subgroup (S1 Table). Despite the critical role TA modules have in regulating the growth of other bacteria, their roles in rickettsial growth and pathogenesis remain unknown. In this study, we functionally characterized two *vapBC* TA modules from *Rh*-B8. We showed that the expression of two VapC toxins in *Rh*-B8 exhibited distinct toxic effects in suppressing bacterial protein synthesis. Our finding highlights the crucial role of the *vapBC*1 in *Rh*-B8's adaption to oxidative stress in human microvascular endothelial cell lines (HMEC-1) during the early infection and establishing infection in a mouse model. Notably, it was discovered that rickettsial VapC2 toxin can reach host cytoplasm and contribute to the pathogenesis of *Rh*-B8 by targeting host rRNAs.

## Results

### *vapBC* TA modules from *Rh*-B8 were activated during the early infection

Most obligate host-associated parasites were found to be resistant to retaining TA modules due to their potential toxic effect on eukaryotes, except for one obligate intracellular pathogen, *Rickettsia* [24]. After searching in the NCBI genome databank, we found that while TA modules are absent from the TG rickettsiae, they are extensively distributed and frequently present in high numbers in the spotted fever group, transitional group, and ancestral group rickettsiae (representative strains from each subgroup were given in S1 Table). Besides, a series of incomplete TA modules were identified in their genome sequences, either with a single antitoxin gene or a toxin gene that prematurely terminated (S1 Table). However, little is known about the characteristics of the remaining intact TA modules in *Rickettsia*.

The genomic analysis showed that all SFG rickettsiae harbor three types of intact TA modules, each with highly conserved amino acid sequences across the SFG subgroup and annotated as VapBC, HicAB, and YefM-YoeB, respectively (S1 Table and Fig 1a). To assess their physiological role, the expression level of each TA module during the infection of *Rh*-B8 in HMEC-1 was evaluated. The results showed that *Rh*-B8 multiplies extremely slowly until 24 hours before entering the fast-growing period (Figs 1b and S1a). The expression of both the *vapB* and *vapC* genes from two *vapBC* modules increased immediately upon infection and remained at a high level until 24 hours post-infection (hpi), followed by a decline to a low level (Fig 1c and 1d). TA modules were typically upregulated to support bacterial survival under stressful environments. The cytopathic alteration seen in the late infection may imply an unfavorable intracellular environment for *Rickettsia*. However, the expression of both *vapBC* modules has remained low from 24 hours post-infection till the final infection (Fig 1c and 1d). Besides, the expression of the *yefM-yoeB* and *hicAB* modules were either continuously kept at a relatively low level or slightly elevated during the late infection (S1b, S1c and S1d Fig). This temporal pattern indicated that *vapBC* TA modules are activated during the early infection and may help overcome initial challenges once *Rh*-B8 invades host cells.

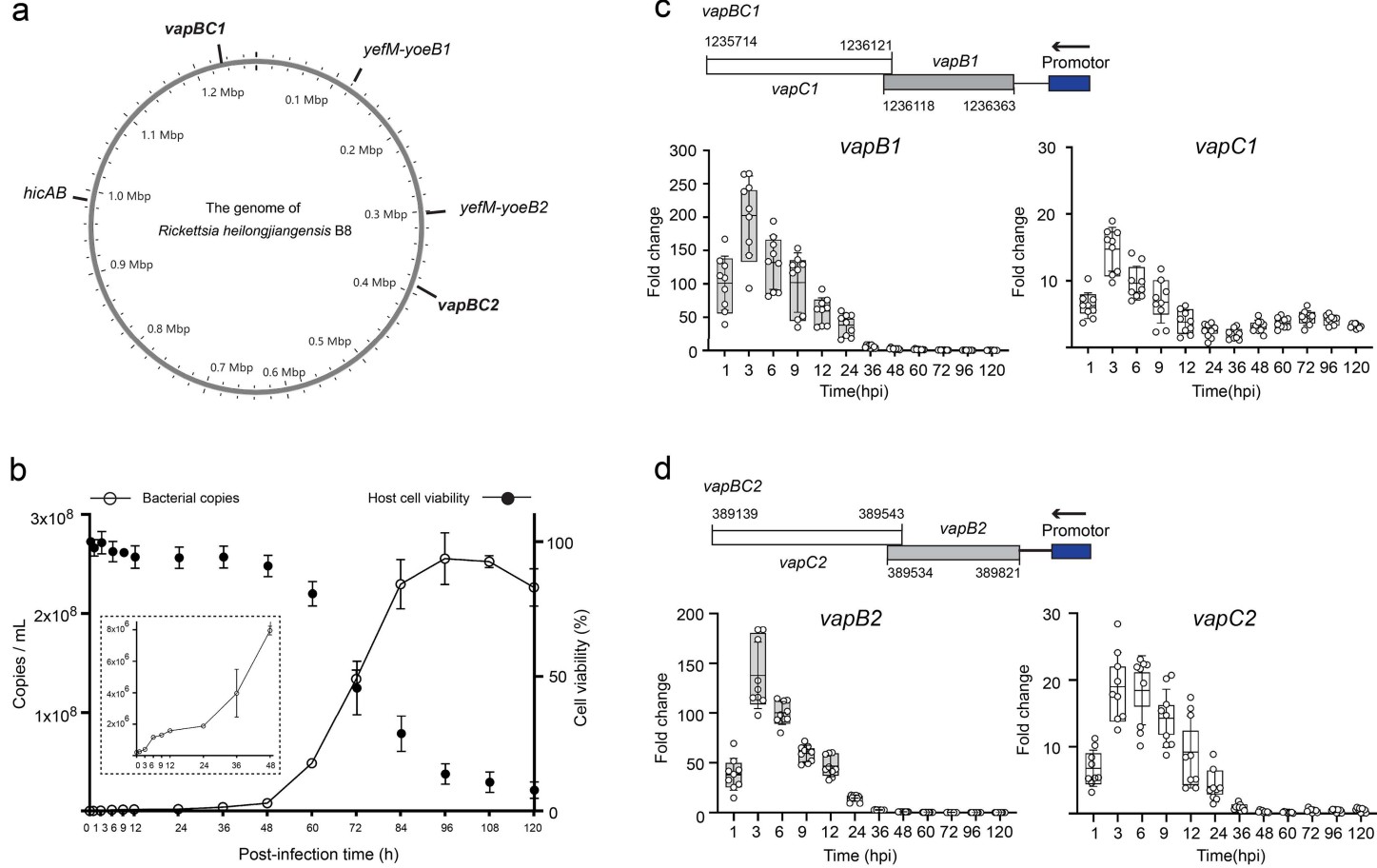

**Fig 1. The transcription of two *Rh*-B8 *vapBC* modules was activated during the early infection.** (a) Chromosomal location of 5 intact TA modules in *Rh*-B8. (b) Growth curves and host cell viability during the infection of *Rh*-B8 in HMEC-1 cell lines (MOI = 0.1). A magnified view of the bacterial growth within the first 48 hours was shown in the inset. (c) and (d) Genomic localization of two *vapBC* TA modules and their transcription level during bacterial growth. Gene transcription levels were measured after normalization to levels of *ompB*. The data was shown as the fold change in comparison to 0 hpi. Data with the mean ± SD are from n = 3 independent experiments, each with three technical replicates.

### *Rh*-B8 VapBCs have distinctive intramolecular interaction patterns

The VapBC complex was previously discovered to form either a biological heterotetramer (VapB$_2$C$_2$) or heterooctamer (VapB$_4$C$_4$). The N-terminal domain of the VapB antitoxin folded as a DNA binding domain to interact with the promoter region of the *vapBC* operon and suppress its own transcription, while the C-terminal tail wrapped around the toxin and closely interacted with the active site to neutralize its activity [25–27]. To understand the mode of interaction within *Rh*-B8 VapBCs, their 3D structures were first computed based on AlphaFold Protein Structure Database [28]. Both VapCs were predicted to be canonical PIN-domain proteins, each containing a conserved catalytic triad of three acidic residues at their active sites [21] (Figs 2a, S2a and S2b). Transcriptional suppression on *vapBC* operon likely occurs through the dimeric, Phd-like DNA-binding domain formed by the N-terminal region of VapB antitoxins [29] (S2c and S2d Fig). Structural predictions revealed that while VapB1 maintains an extended C-terminal tail typical of characterized VapB antitoxins, VapB2 possesses a unique C-terminal architecture featuring two α-helices followed by a short tail segment (Figs 2a, S2a and S2b) [25,27,30]. To characterize VapB-VapC interactions, we performed GST pull-down assays using purified wild-type

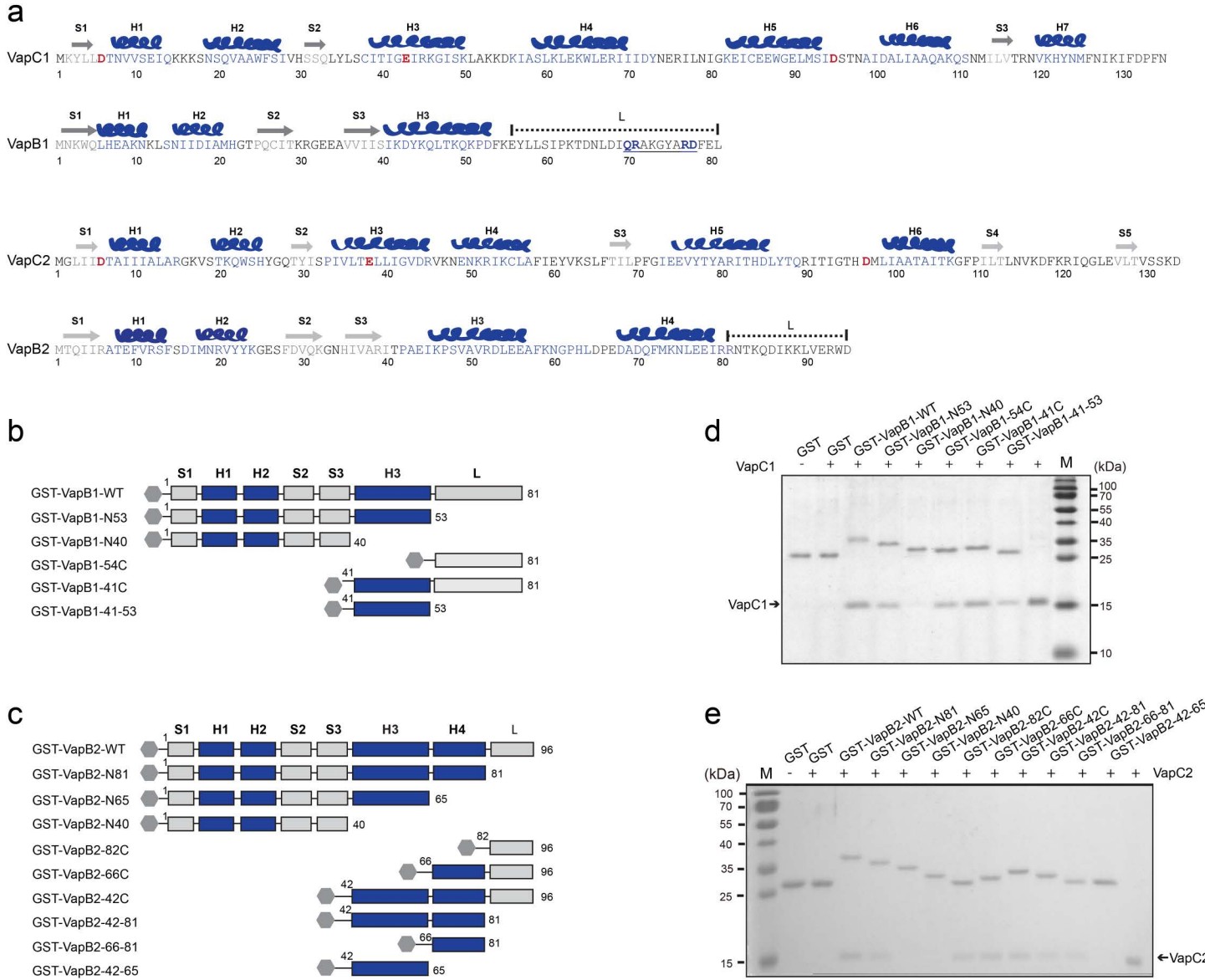

**Fig 2. The toxin-antitoxin interaction patterns of *Rh*-B8 VapBC modules.** (a) The amino acid sequences of *Rh*-B8 VapBC modules. The secondary structure elements were annotated based on the predicted 3D structure, with β-sheets labeled as 'S', α-helices as 'H', and irregular loop regions as 'L'. Conserved acidic residues of the "catalytic triad" in VapC were indicated in red. The region of the "pseudo-palindromic" sequence in VapB1 was underlined. The disordered C-terminus of VapB antitoxin was indicated using dashed lines. (b) and (c) Schematic representation of wild and truncated VapB1 and VapB2 fused to an N-terminal GST tag. (d) VapC1 binding to GST-VapB1. Lane 1, GST without VapC1; lane 2, GST with VapC1; lanes 3–8, GST fused to wild-type and truncated mutants of VapB1 with VapC1, as indicated above each lane; lane 9, VapC1. (e) VapC2 binding to GST-VapB2. Lane 2, GST without VapC2; lane 3, GST with VapC2; lanes 4–13, GST fused to wild-type and truncated mutants of VapB2 with VapC2, as indicated above each lane; lane 14, VapC2.

VapCs and GST-fused VapB truncation mutants (Fig 2b and 2c). The results demonstrated that the C-terminal regions of both VapBs mediate binding to their cognate VapC toxins (Figs 2d, 2e, S2e and S2f). VapB1 was found to require both its H3 helix and extended C-terminal tail for interaction (Figs 2d and S2e), similar to the VapB-VapC binding mode observed in *Caulobacter crescentus* [27]. Notably, VapB1's C-terminal tail contains a pseudo-palindromic motif (Fig 2a), suggesting

its ability to simultaneously block both active sites in the VapC1 dimer (S2g Fig) [27]. In contrast, VapB2 only requires its H4 helix and short C-terminal tail for VapC2 binding (Figs 2e and S2f). This configuration closely resembles the FitA-FitB interface in *Neisseria gonorrhoeae* [31], where the L-shaped structural motif spanning from the H4 helix to the C-terminus mediates direct binding to VapC2 (S2b and S2h Fig). Together, these findings indicate that the two *Rh*-B8 VapBC complexes may employ distinct molecular strategies for antitoxin-toxin recognition.

## Lon protease triggers *vapBC* activation under host oxidative stress

To assess environmental factors responsible for the activation of *vapBC* modules, we examined the transcript level of both *vapBC* modules in response to various stress conditions or drugs. In this study, the *vapB* genes were examined as representatives of their respective *vapBC* operons, with their transcription profiles monitored over a 24-hour period following stimulation initiated at 12 hours post-infection (hpi). The results showed that the transcript level for both *vapBC* modules was unaffected by either nutrition limitation (NL) or exposure to nitrosative stress ($NO_2^-$). The transcript level decreases in both cases, identical to that in normal infection (Fig 3a and 3b). Transcription of *vapBC* modules also declined as in normal infection following exposure to antibiotics (chloramphenicol and rifampin), but there was a noticeable rise after 24 hours of stimulation (Fig 3a and 3b). However, when exposed to $H_2O_2$, *vapBC*1 transcription increased rapidly, and *vapBC*2 transcription stayed high rather than declining as it did in unstimulated condition (Fig 3a and 3b). We then aim to examine how $H_2O_2$ exposure affects the transcription of *vapBC* modules at different stages of infection. After being exposed to $H_2O_2$ at 24 hpi, both *vapBC* transcript levels were also strongly elevated in the following 12 hours before a significant decline (Fig 3c and 3d). However, their transcription remains unaltered when exposed to $H_2O_2$ initiated at 48 or 72 hpi (Fig 3c and 3d). Furthermore, we assessed the transcription levels of *vapB1* and *vapB2* during early *Rh*-B8 infection in *Keap1*-knockout HMEC-1 cells, which exhibit impaired ROS production due to constitutive antioxidant enzyme overexpression [32]. The results demonstrated significantly reduced transcriptional upregulation of both *vapBC* modules compared to infection in normal HMEC-1 cells (Fig 3e and 3f). Together, these findings suggested that the upregulation of *Rh*-B8's *vapBC* modules is induced by host oxidative stress exposure during early infection.

Vascular cells were discovered to produce ROS that are essential in protecting against intracellular pathogens. As a result, the oxidative stress environment may represent a real threat once *Rickettsia* enters host cells. Therefore, we assessed the oxidative state of HMEC-1 cell lines following *Rh*-B8 infection by measuring the quantity of free ROS. The results demonstrated a substantial increase of free ROS at the initial infection and maintained at a high level until 24 hpi before significantly declining (Fig 3g). These findings suggest that *Rh*-B8 triggers host ROS production during early infection stages. Intriguingly, the metabolic study on *Rh*-B8 infected HMEC-1 revealed that intracellular glutathione (GSH) in HMEC-1 was significantly accumulated since 24 hpi (Fig 3h), which provides a clue to elucidate the ROS fluctuation observed in the host, given that GSH is important in maintaining redox equilibrium by eliminating oxidants [33]. Thus, the reduced GSH concentration at each selected time point was quantified. The result showed that GSH started to be generated and accumulated upon infection, appeared most abundant at 24 hpi, and remained high until the late infection (Fig 3i). We next examined the response from the oxidant-scavenging enzyme system in HMEC-1, including glutathione peroxidase (GSH-Px), glutathione reductase (GSR), superoxide dismutase (SOD), and catalase. The result demonstrated an increase to various extent for these enzymes (Fig 3j). These results suggested that *Rh*-B8 infection induces ROS accumulation in HMEC-1 cells, which might be neutralized by an antioxidant response from host cells to prevent severe oxidative self-damage. Furthermore, these findings can also elucidate why, in the later stage of infection (48 or 72 hpi), exposure to $H_2O_2$ failed to upregulate *vapBC* transcription (Fig 3c and 3d). Collectively, these results support the notion that *vapBC* modules play an important role in supporting *Rickettsia*'s adaptation when confronted with a host oxidative challenge during early infection.

The release of functional toxin depends on the breakdown of antitoxin by endogenous ATP-dependent proteases [34,35]. To assess the mode of *Rh*-B8 VapB degradation, we analyzed the transcript level of protease Lon and

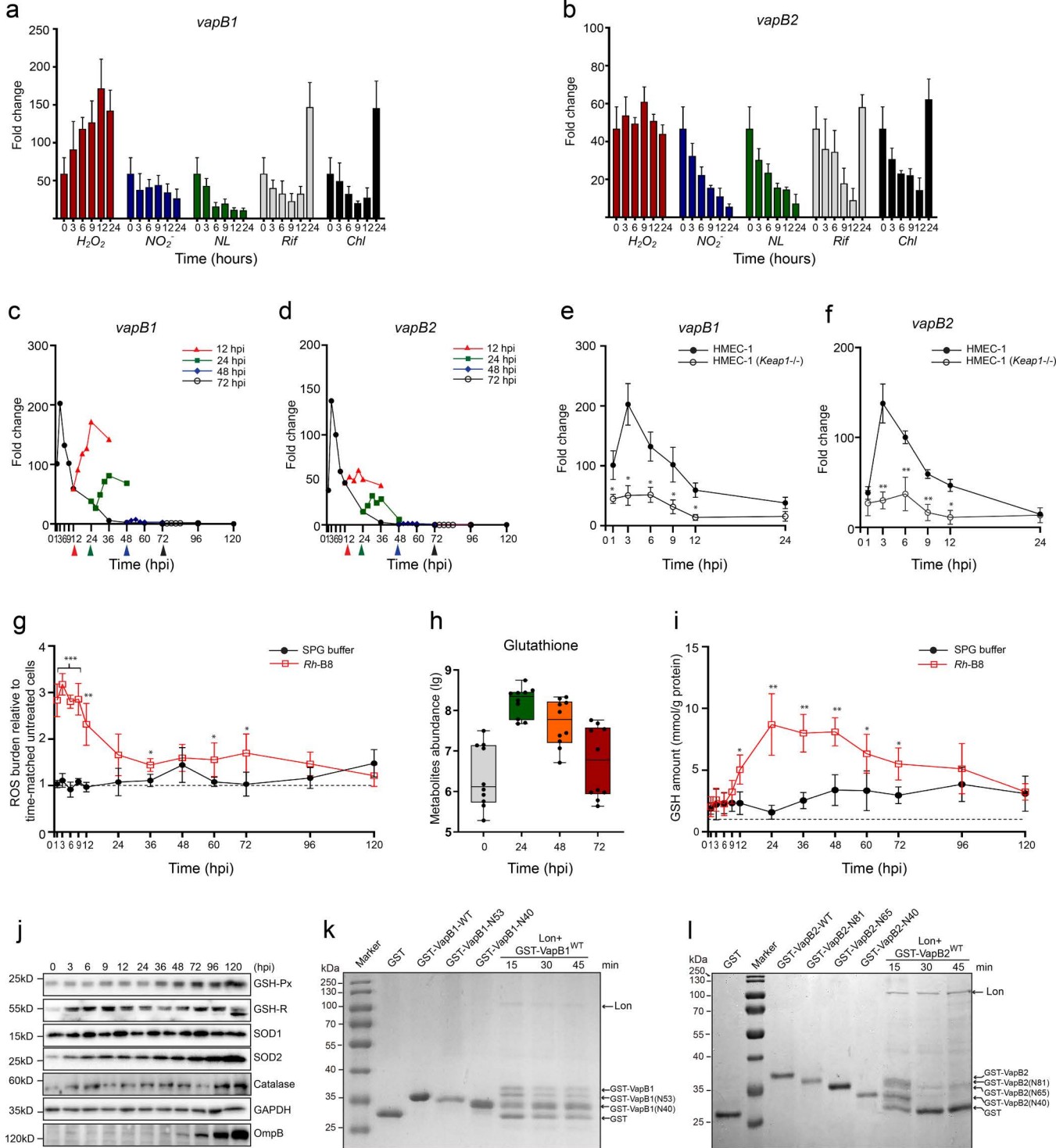

**Fig 3. Oxidative stress activates *vapBC* modules.** (a) and (b) Transcripts level of *vapBC* modules upon stress stimulation at 12 hpi, for 3, 6, 9, 12, and 24 hours, respectively. The relative transcription level was quantified upon exposure to oxidative stress ($H_2O_2$), nitrosative stress ($NO_2^-$), nutrient limitation (NL), Rifampin (Rif), or Chloramphenicol (Chl). Gene transcription levels were measured after normalization to levels of *ompB*. The data was shown as the fold change in comparison to 0 hpi. Bar represents the mean ± SD from n = 3 independent experiments. (c) and (d) Transcripts level of *vapB* upon $H_2O_2$ exposure for 3, 6, 9, 12, and 24 hours, at 12, 24, 48, or 72 hpi, respectively. Data at each time point represents the mean value from

n = 3 independent experiments, each with three technical replicates. (e) and (f) Transcript levels of *vapBC* modules during early infection in wild-type HMEC-1 cells versus *Keap1*-deficient (*Keap1*-/-) cells with impaired ROS production. Data represent mean ± SD from three independent experiments. Statistical significance (*p < 0.1, **p < 0.01) was determined by unpaired t-test comparing *Keap1*-/- to wild-type HMEC-1 controls. (g) Reactive oxygen species (ROS) abundance in HMEC-1 cell lines upon *Rh*-B8 infection at an MOI of 0.1. ROS levels were measured and expressed as ratios relative to time-matched untreated cells. Data with the mean ± SD are from n = 3 independent experiments (* p < 0.1, ** p < 0.01, *** p < 0.001 relative to time-matched SPG-treated sample, p-value calculated using an unpaired t-test (two-tailed)). (h) GSH abundance in HMEC-1 upon *Rh*-B8 infection from metabolism analysis (n = 10 independent experiments). (i) GSH amounts in HMEC-1 upon SPG treatment and *Rh*-B8 infection were quantified using the DTNB method. Data with the mean ± SD are from n = 3 independent experiments (* p < 0.1, ** p < 0.01, *** p < 0.001 relative to time-matched SPG-treated sample, p-value calculated using an unpaired t-test (two-tailed)). (j) The protein level of host antioxidant enzymes, including glutathione peroxidase (GSH-Px), glutathione reductase (GSR), superoxide dismutase (SOD) 1/2, and catalase from HMEC-1 at indicated time points upon infection of *Rh*-B8. OmpB from *Rh*-B8 was used to indicate bacterial growth. (k) and (l) *In vitro* cleavage assay on VapB antitoxins by purified Lon protease. Lanes 2-5 in (k) and lanes 1, 3-6 in (l) represent GST and GST-fused VapB truncates used as indicators for the cleavage products, which were shown in lanes 6-8 of (k) for VapB1, lanes 7-9 of (l) for VapB2.

caseinolytic or caspase-3-like protease (Clp)X/P coding genes present in *Rh*-B8 genome. It's intriguing to note that during the early infection when intracellular ROS was accumulating, the elevation of *lon* transcript level corresponds with *vapBC* operons (S3a Fig), but the transcription of *clpX* was unaltered (S3b Fig). Lon protease was widely distributed throughout all spheres of life and was shown to mediate broad adaption reactions to a variety of survival difficulties [36,37]. Here we performed *in vitro* degradation assay using GST-tagged VapBs and purified *Rh*-B8 Lon protease. The results demonstrate that the C-terminal interaction domains of both VapB antitoxins (residues 54–81 in VapB1 and 82–96 in VapB2), which are essential for binding their cognate VapC toxins, show significantly higher susceptibility to Lon protease degradation (Fig 3k and 3l). Our proteolysis analysis revealed distinct degradation profiles between the two antitoxins: while VapB1 showed partial degradation with preservation of most N-terminal residues (Fig 3k), VapB2 underwent more complete proteolytic processing under identical conditions (Fig 3l). Collectively, these findings demonstrate that rickettsial VapB antitoxins are vulnerable to Lon-mediated proteolysis during host oxidative stress conditions. This regulated degradation leads to subsequent liberation of active VapC toxins, suggesting a plausible stress-response mechanism in *Rickettsia*.

### *vapBC1* deficient strain is more susceptible to oxidative stress *in vitro* and attenuated for growth in mice

To understand the physiological role of *vapBC* modules, we constructed *vapC1* and *vapC2* mutant strains of *Rh*-B8 by targeting the insertion of intronic RNA using the LtrA group II intron retrohoming system as previously described [38]. The disruption of *vapC1* and *vapC2* was confirmed using PCR to amplify full-length *vapC* genes. PCR products with apparent larger size were sent for sequencing, and the results confirmed the insertion with in-frame stop codons after nucleotide 303 for *vapC1* (*vapC1::int303*) and nucleotide 240 for *vapC2* (*vapC2::int240*) mutant strains, respectively (S4 a and S4b Fig). Because the *vapC1::int303* and *vapC2::int240* mutants were identified due to smaller plaque phenotypes during individual plaque selection on Vero cell monolayers, we next compared plaque area for WT and mutant strains in HMEC-1 cell lines. In comparison with WT, both mutant strains, especially the *vapC1::int303* strain, produced significantly smaller plaques (Fig 4a and 4b). This finding suggested that both *vapBC* modules contribute to *Rh*-B8's intracellular multiplication. Next, the growth curves of *Rh*-B8 in HMEC-1 and Vero cells were assessed using the plaque-forming units (PFU) method. In both cell lines, the mutant strains' rickettsial replication was impaired in comparison to WT, and it was shown that the *vapC1::int303* strain's proliferation in HMEC-1 was significantly reduced (Fig 4c and 4d). These results showed that *Rh*-B8's reproduction in HMEC-1 cell lines depends more on *vapBC*1 even though both *vapBC* modules support normal rickettsial proliferation.

Our findings demonstrated that transcript upregulation of both *vapBC* modules is induced by host-derived ROS accumulation following *Rh*-B8 infection. To further elucidate VapC's role in oxidative stress response, we conducted comparative growth kinetics analyses using Δ*vapC* mutant strains in *Keap1*-knockout HMEC-1 cells. Both wild-type and VapC-deficient strains exhibited increased replication rates during infection, accompanied by accelerated host cell damage,

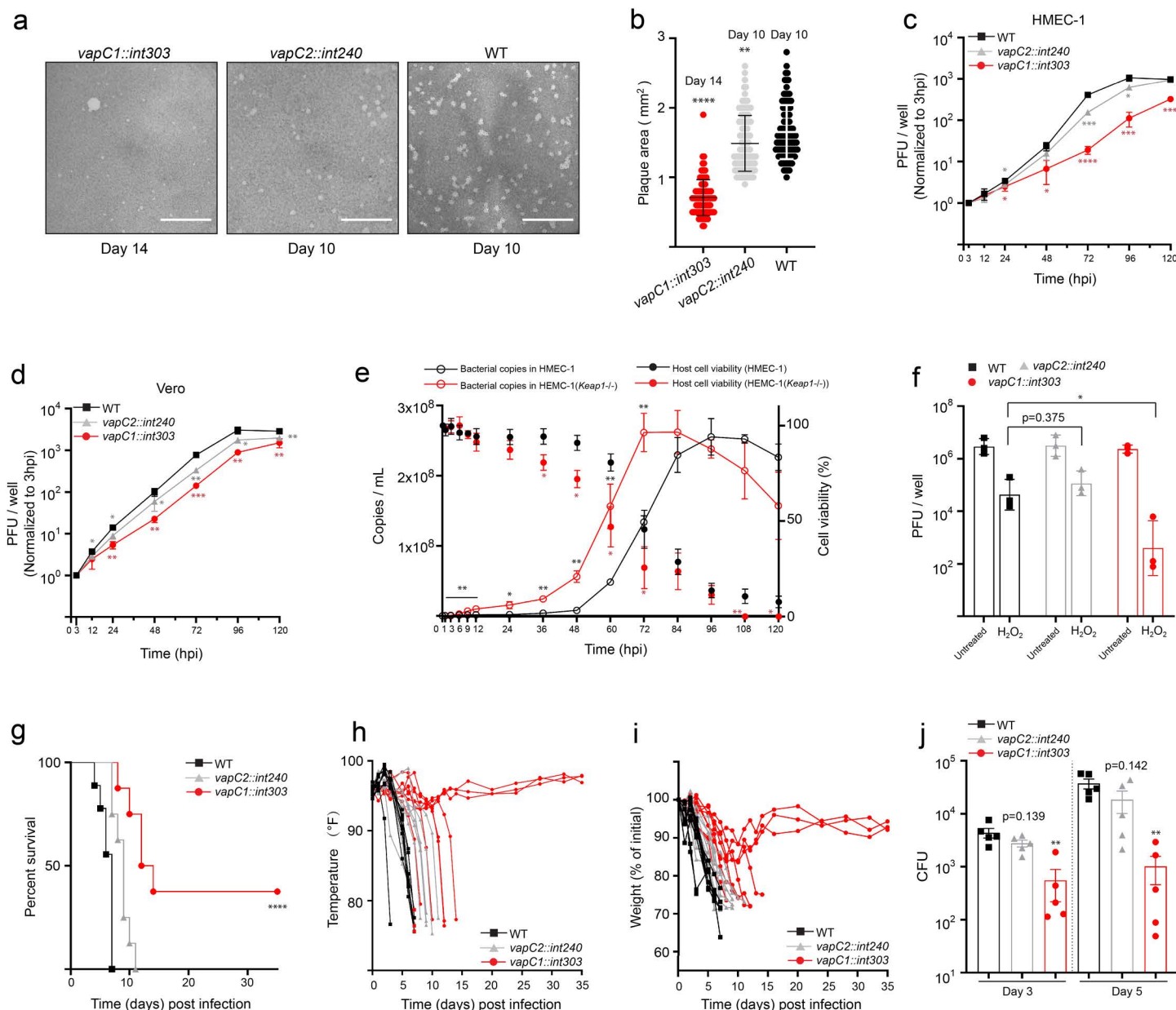

**Fig 4. *vapBC* modules are crucial for the infection and multiplication of *Rh*-B8 in host cells and mouse models.** (a) Representative images of plaques stained with neutral red at 10 days post-infection (dpi) (WT and *vapC2::int240* strain) and 14 dpi (*vapC1::int303* strain). Scale bar 10 mm. (b) Plaque areas in HMEC-1 monolayer infected with WT (10 dpi), *vapC1::int303* strain (14 dpi), and *vapC2::int240* strain (10 dpi) (n = 2 independent experiments; WT n = 97 plaques, *vapC1::int303* strain n = 90 plaques, *vapC2::int240* strain n = 106 plaques). All data points were shown, with the bar representing mean ± SD (* p < 0.1, ** p < 0.01, *** p < 0.001, **** p < 0.0001 relative to WT strain, p-value calculated using an unpaired t-test (two-tailed)). (c) and (d) Bacterial abundance of WT, *vapC1::int303*, and *vapC2::int240* strains in (c) HMEC-1 and (d) Vero cell lines (n = 3 independent experiments), MOI = 0.1. Data are mean values with ± SD; * p < 0.1, ** p < 0.01, *** p < 0.001, **** p < 0.0001 relative to WT strain, p-values were calculated using two-tailed unpaired t-test. (e) Growth kinetics and host cell viability during infection with WT *Rh*-B8 strain in HMEC-1 and HMEC-1 (*Keap1-/-*) cells at MOI = 0.1. Data represent mean ± SD from three independent experiments. Statistical significance (*p < 0.1, **p < 0.01) was determined by two-tailed unpaired t-test comparing *Keap1$^{-/-}$* to wild-type HMEC-1 cells. (f) Susceptibility of WT, *vapC1::int303,* and *vapC2::int240* strains to H$_2$O$_2$ exposure (100 µM) for 24 hours, MOI = 5.0. Data are mean values with ± SD obtained from n = 3 independent experiments, each with three technical replicates. (g) Survival of *Ifnar*1$^{-/-}$ mice infected with WT, *vapC1::int303* and *vapC2::int240* strain. Data were analyzed using a log-rank (Mantel-Cox) test, **** p < 0.0001. (h) Temperature changes over time in *Ifnar*1$^{-/-}$ mice infected with WT, *vapC1::int303,* and *vapC2::int240* strain; data from individual mice were shown (n = 8 mice for WT, n = 9 mice for *vapC2::int240*, n = 8 mice for *vapC1::int303,* n = 2 independent experiments). (i) Weight change over time in *Ifnar*1$^{-/-}$ mice

infected with WT, *vapC1::int303,* and *vapC2::int240* strain, data from individual mice were expressed as percent change from initial weight. (j) The bacterial loads were determined using qPCR with collected spleen homogenates after intravenous infection. The data in these panels are mean±SD of log10 CFU obtained from 5 infected mice per group. ** p<0.01 relative to WT strain, p-values are calculated using an unpaired t-test (two-tailed).

likely attributable to faster bacterial proliferation (Figs 4e, S4c and S4d). Notably, the VapC-deficient strains reached peak bacterial loads comparable to those of the wild-type (Figs 4f, S4c and S4d), indicating that VapC activity may become dispensable for proliferation under oxidative stress-deficient conditions. Furthermore, these findings suggest that *Rh*-B8 VapC toxins may play a regulatory role in rickettsial proliferation during oxidative stress. Next, we investigated and compared the susceptibility of strains WT, *vapC1::int303*, and *vapC2::int240* mutants after exposure to an oxidative stress environment. In this experiment, we quantitatively assessed viable bacterial counts following 24-hour $H_2O_2$ exposure, with treatment initiated at 1-hour post-infection (hpi) using a multiplicity of infection (MOI) of 5.0. It was found that the *vapC1::int303* mutant strain was more susceptible to killing by $H_2O_2$ in comparison to WT and *vapC2::int240* (Fig 4f). Collectively, these findings indicated that the capability of the *vapBC*1 module in responding to oxidative stress is crucial for *Rh*-B8's intracellular growth.

To investigate the contribution of *vapBC* modules to rickettsial virulence *in vivo*, C57 mice lacking IFN-I receptors (*Ifnar1*⁻/⁻ knock-out mice) that succumb to wild-type *Rh*-B8 infection were used in mouse model research [39]. Mice showed a rapid drop in body temperature and weight following intravenous infection with 1.0 x 10$^5$ PFU WT bacteria and did not survive beyond day 7 (Fig 4g, 4h and 4i). The majority of the *vapC2::int240* mutant infected mice did not survive past day 10, exhibiting a similar pattern of mortality when compared to WT (Fig 4g). On the other hand, several mice infected by *vapC1::int303* mutant strain showed an initial decline in weight and body temperature followed by recovery within 2 weeks and survived until the end of the experiment (day 35). The body temperature and weight of other mice gradually decreased, and half of them survived for more than ten days (Fig 4g, 4h and 4i). Furthermore, gross pathology on mice that died of infection by either WT or mutant strains revealed the most apparent pathological changes in spleens (S5a Fig). In agreement, large amounts of *Rickettsia* in spleens could be observed via immunohistochemical staining analysis (S5b Fig). We next examined the development of spleen bacterial burdens in mice upon infection by WT and mutant strains. We observed that rickettsial loads were significantly attenuated in mouse spleen tissues upon infection by *vapC1::int303* mutant strain compared with that infected by either WT or *vapC2::int240* mutant strain at day 3 and 5, respectively (Fig 4j). Together, our data indicated that *vapBC*2 module may have a less important role during *Rh*-B8's reproduction, while *Rh*-B8 missing intact *vapBC*1 is attenuated for growth and pathogenicity in vein-infected animals.

### *Rh*-B8 VapCs perform distinct toxic effects on bacterial growth

To assess the function of VapC toxins from *Rh*-B8, VapC coding genes were first introduced downstream of a plasmid-encoded arabinose inducible promoter and expressed in *E. coli* (S6a Fig). Despite the induction of VapC2 has no effect on bacterial growth, induction of VapC1 causes significant growth suppression (Fig 5a and 5b), and co-expression of the cognate VapB1 restored bacterial growth (Figs 5a, 5b, and S5a). In agreement, the [³⁵S]methionine incorporation assay demonstrated a substantial reduction in the global rate of translation following VapC1 induction and the analogous experiments on [³H]uracil incorporation showed that transcription was not affected (Fig 5c and 5d). In contrast, neither translation nor transcription inhibition were observed upon VapC2 induction (Fig 5c and 5d). These results showed that VapC1 primarily inhibited protein synthesis to cause bacterial growth arrest which could be reversed by VapB1 antitoxin, whereas VapC2 had no negative effects on *E. coli*.

To evaluate the effect of free VapC toxins in *Rickettsia*, the VapC coding genes were introduced downstream of a *rompB* promoter in a rickettsial shuttle plasmid as previously described [40] (S6b Fig). After transformation, *Rh*-B8 overexpressing VapC1$^{D6A}$ (a predicted catalytically inactive VapC1, *Rh*-B8$^{vapC1(D6A)+}$), wild-type VapC2 (*Rh*-B8$^{vapC2+}$), and VapC2$^{D6A}$

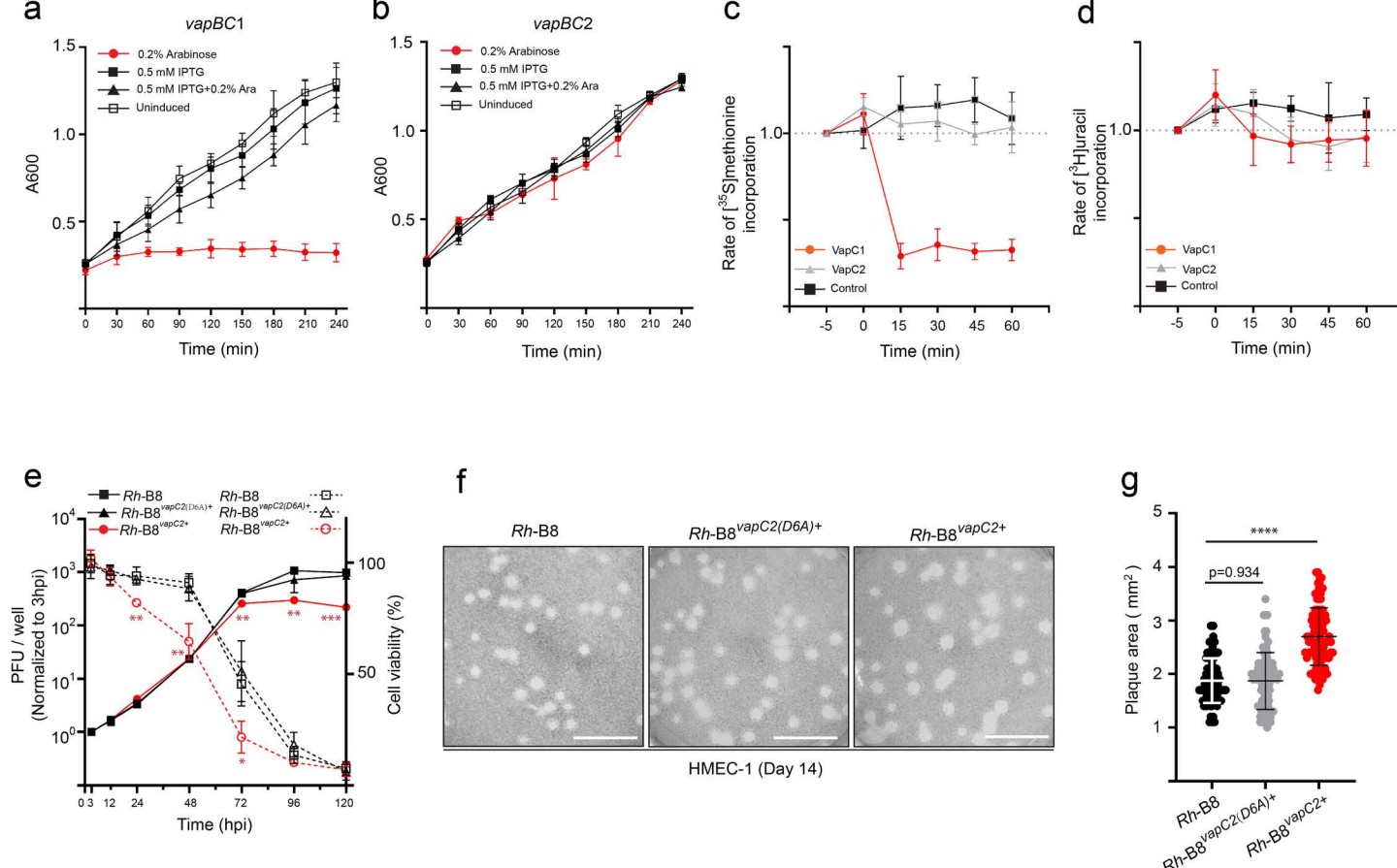

**Fig 5. Effect of *Rh*-B8 VapCs' expression on bacterial growth.** (a) The growth rates of *E. coli* K-12 cells carrying pBAD-VapC1 and pACYCDuet-1-VapB1 under different induction conditions. Cultures were grown in the LB medium at 37 °C. At time zero, pBAD-VapC1 was induced with 0.2% L-arabinose, and pACYCDuet-1-VapB1 was induced with 0.5 mM IPTG. (b) The growth rates of *E. coli* carrying pBAD-VapC2 and pACYCDuet-1-VapB2 under different induction conditions were compared as described in (a). (c) and (d) Rates of translation (c) and transcription (d) of *E. coli* K-12 cells carrying the pBAD empty vector, pBAD-VapC1, or pBAD-VapC2. *E. coli* K-12 were grown exponentially in M9 minimal medium at 37 °C. Samples at indicated time points were collected and pulsed with [$^{35}$S]methionine (c) or [$^3$H]uracil (d). At time zero, culture was induced with 0.2% arabinose. Data were normalized according to the $A_{600}$ value at each time point and plotted relative to the value at -5 min (rates of translation or transcription were set to 100%). The data shown were averages of three independent experiments with resulting ±SD. (e) The bacterial abundance of WT (square) and strain expressing wild-type VapC2 (circle) or inactive mutant VapC2$^{D6A}$ (triangle) in HMEC-1 cell lines (solid lines), and the corresponding host cell viability (dashed lines). Data shown are the mean values from 3 independent experiments with resulting ±SD; * p<0.1, ** p<0.01, *** p<0.001 relative to WT strain, p-values were calculated using an unpaired t-test. (f) Representative images of plaques formed by wild-type strain *Rh*-B8 and strain expressing VapC2 or VapC2$^{D6A}$. Scale bar 10 mm. (g) Plaque areas in HMEC-1 cells infected with WT, VapC2$^{WT}$, and VapC2$^{D6A}$ expressing strain (n=2 independent experiments; WT n=95 plaques, a strain expressing VapC2$^{WT}$ n=98 plaques, a strain expressing VapC2$^{D6A}$ n=90 plaques). All data points were shown with the bar representing mean±SD (**** p<0.0001 relative to WT strain, p-value calculated using an unpaired t-test).

(a predicted catalytically inactive VapC2, *Rh*-B8$^{vapC2(D6A)+}$) were successfully selected using the plaque method supplemented with rifampin (S6c Fig). Attempts to select colonies expressing wild-type VapC1 were unsuccessful, indicating that overexpression of wild-type VapC1 might also cause growth arrest of *Rickettsia*, as seen for *E. coli*. To determine whether VapC2 has a function during bacterial replication, growth curves measuring plaque-forming units (PFU) were performed with an HMEC-1 monolayer. It was shown that bacterial multiplication was impaired for *Rh*-B8$^{vapC2+}$ after 72 hpi but not for *Rh*-B8$^{vapC2(D6A)+}$. Besides, host cell viability following *Rh*-B8$^{vapC2+}$ infection dramatically declined since 24 hpi rather than

from 72 hpi upon infection by strain WT or $Rh$-B8$^{vapC2(D6A)+}$ (Fig 5e). This result suggested that overexpression of VapC2 in $Rh$-B8 may promote host cell damage, thereby inhibiting late-stage bacterial reproduction due to rapid exhaustion of host cells. To identify this hypothesis, we next compared plaque area for strains WT, $Rh$-B8$^{vapC2(D6A)+}$, and $Rh$-B8$^{vapC2+}$. The result showed that $Rh$-B8$^{vapC2+}$ displayed comparatively larger plaques in comparison to WT and strains expressing inactive Vap-C2$^{D6A}$, indicating that the strain overexpressing VapC2 would be more virulent for host cells (Fig 5f and 5g). Based on the above results, it appears that VapC2 from $Rh$-B8 was detrimental to host cells but not $Rickettsia$ itself.

### VapC1 slows bacterial growth by specifically cleaving tRNA$^{fMet}$ in $Rh$-B8

To investigate the cellular target of $Rh$-B8 VapCs, we first assessed the total RNA isolated from $E. coli$ harboring vectors expressing wild-type VapC1/2. Here the VapC20 from $M. tuberculosis$, which specifically cleaves 23S rRNA and releases a ~ 250 nt RNA fragment, was used as a control to verify toxin expression [41]. The results showed that expression of wild-type VapC1 resulted in a ~ 50 nt RNA fragment, and no cleavage was observed for VapC1$^{D6A}$ (Fig 6a). Moreover, there was no discernible specific RNA breakage following the induction of wild-type VapC2 (S7a Fig). Additionally, to rule out the potential of unspecific breakage on rRNA that was not visible with EB staining, northern blotting was performed using specific probes that target the 5' or 3' sequence of 23S rRNA and 16S rRNA from $E. coli$ (S7b and S7c Fig). These results also support our above conclusion that VapC2 has no detrimental impact on $Rh$-B8 itself.

To identify the cleavage products of VapC1, the resulting small RNA fragments were purified, ligated with adaptors, and sent for sequencing [42] (S7d Fig). The result identified a range of tRNA fragments, which could be the impacts of VapC's overexpression. The most abundant were from the cleavage products of the tRNA$^{fMet}$, tRNA$^{Asp}$, tRNA$^{Thr}$, and tRNA$^{Gly}$, respectively (Figs 6b, 6c, and S7e). It was found that all putative tRNA targets had cleavage sites introduced by VapC1 in the anticodon stem-loop region (Figs 6d and S7f). To further identify VapC1's targets in $Rickettsia$, the $in vitro$ cleavage assay was performed with the purified wild-type VapC1 and total small RNA preparations of $Rh$-B8. A cleavage product that corresponds to the anticipated tRNA fragment could be detected (Fig 6e). Northern blotting was then used to confirm the cleavage products using probes specific to each potential tRNA target. Our results demonstrated that the RNA fragment originates specifically from tRNA$^{fMet}$ cleavage, rather than other tRNAs in $Rh$-B8. This indicates that free VapC1 inhibits $Rh$-B8 growth by targeting the anti-codon loop region of the initiator tRNA$^{fMet}$ (Fig 6f and 6g). To further elucidate the mechanism by which $Rh$-B8 regulates growth, we quantified initiator tRNA$^{fMet}$ abundance in $Rh$-B8 during infection. The results revealed extensive tRNA$^{fMet}$ degradation within the first 24 hours post-infection, followed by full restoration (Fig 6h–j). These findings align precisely with $Rh$-B8's growth kinetics, supporting our conclusion that the delayed early-phase growth results from VapC1-mediated depletion of initiator tRNA$^{fMet}$.

### $Rh$-B8 VapC2 induce host cell arrest by non-specific cleavage on host ribosome RNAs

We have shown that $Rh$-B8 strain that expresses VapC2 possesses a toxic effect on the host rather than $Rickettsia$ itself. To identify this notion, we first wish to investigate whether rickettsial VapCs could be exposed to host cytoplasm during infection. $Rh$-B8 was transformed with plasmids encoding GSK (glycogen synthase kinase)-tagged VapC1$^{D6A}$, VapC2$^{WT}$, or GFP proteins. The GSK tag is a short phosphorylatable peptide upon exposure to host cytoplasmic Ser/Thr kinases [43]. It was found that GSK-VapC1$^{D6A}$ and GSK-VapC2$^{WT}$ were both expressed and phosphorylated during infection, and GSK-GFP was expressed but not phosphorylated (Fig 7a). The result indicated that both rickettsial VapC toxins were capable of being transported and exposed to host cell cytosol. Next, to investigate the impact of VapCs on human cells, GFP proteins were first co-expressed with either the full-length wild-type VapC toxins or inactive mutants in HEK293 cells. It was found that both wild-type VapC toxins promote cytopathic effects, and nearly no co-expressed GFP could be visualized (Figs 7b and S8a). Besides, western blotting revealed minor levels of toxin proteins; however, VapC2 appeared to be more effective in inhibiting the synthesis of proteins in human cells, since much less VapC2 than VapC1 was detected (Fig 7c).

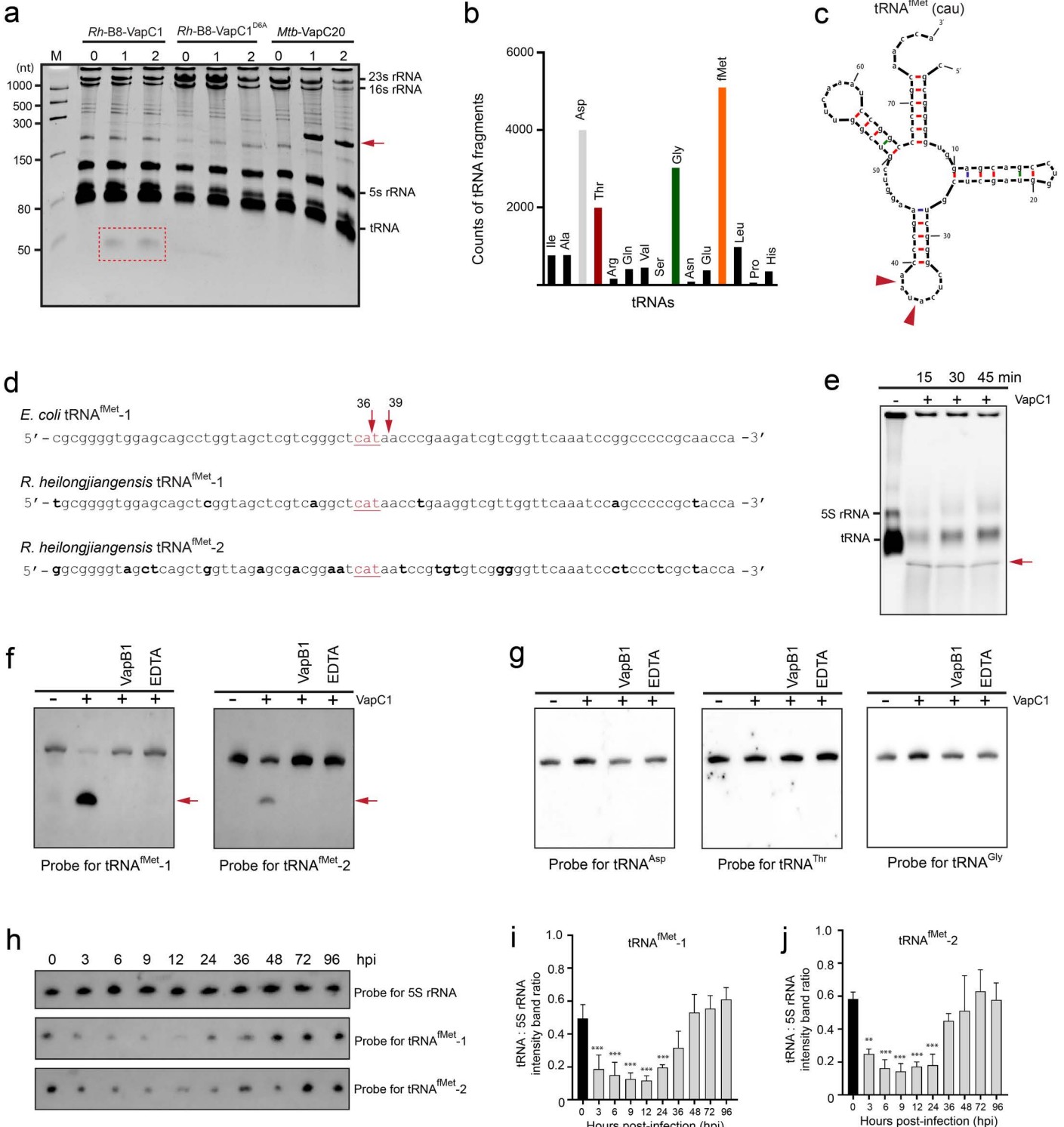

**Fig 6. VapC1 cleaves the anti-codon loop of tRNA^fMet in *Rh*-B8.** (a) Analysis of total RNA from *E. coli* upon induction of VapC1. *E. coli* harboring pBAD-VapC1 and pBAD-VapC1^D6A were grown in the LB medium, and transcription was induced at time zero by adding L-arabinose (0.2%). VapC20 from *M. tuberculosis* (*Mtb*-VapC20) was used as a control. Cell samples were collected at the indicated time points (min). Total RNA extracted from the samples was separated on a 6% denaturing polyacrylamide gel and visualized by ethidium bromide staining. The cleavage products by wild-type VapC1

were indicated with a dashed box, and the *Mtb*-VapC20 cleavage products were indicated with an arrow. (b) Counts of tRNA fragments identified from RNA-seq. (c) Secondary structure diagrams of tRNA^fMet identified as potential VapC cleavage products. The arrows indicate the site of cleavage based on sequencing results. (d) Sequence alignment of *E. coli* tRNA^fMet and *Rh*-B8 tRNA^fMet-1 and tRNA^fMet-2. The anticodon sequences were underlined. The different sequences of *Rh*-B8 tRNA^fMet were shown in bold font. (e) Analysis of total small RNA from *Rh*-B8 after incubation with purified VapC1. VapC1 was inactivated by phenol/chloroform extraction and 1 μg of extracted RNA was detected on an 8% denaturing polyacrylamide gel and visualized by ethidium bromide staining. (f, g) Northern blot analysis of VapC1-mediated tRNA cleavage, as outlined in (e). Probes were specific to (f) tRNA^fMet-1 and tRNA^fMet-2, and (g) tRNA^Asp, tRNA^Thr and tRNA^Gly. Control reactions included excess purified VapB1 antitoxin (lane 3) and EDTA (lane 4). Cleavage products are marked with arrows. (h) Northern blot analysis was performed on total small RNA extracted from *Rh*-B8 at selected post-infection time points using probes specific for tRNA^fMet-1 and tRNA^fMet-2. 5S rRNA served as the loading control. (i) Densitometric quantification of target tRNA bands shown in (h) normalized to 5S rRNA levels (mean±SD; *p<0.1, **p<0.01, ***p<0.001 vs. 0 hpi; unpaired t-test).

There was no co-expressed GFP could be detected in both cases (Fig 7c). In agreement, cell viability assay showed that both VapC toxins were detrimental to human cells, with VapC2 having a more deleterious effect than VapC1 (Fig 7d). The expression of inactive VapC mutants had no negative effects on either protein expression or cell viability (Fig 7b–d).

We next investigated whether tRNA cleavage mediates eukaryotic cell arrest, analogous to the mechanism observed with VapC1 in bacterial systems. Total RNA from HEK293 cells expressing wild-type VapC toxins were isolated and subjected to an agarose gel. The results showed that neither VapC1 nor VapC2 expression in human cells produced distinctive tRNA fragments or decreased the quantity of small RNAs, yet the degradation of 18S and 28S rRNA was noticeable (Fig 7e and 7f). *In vitro* cleavage assay was performed with purified wild-type VapCs and the total RNA isolated from HMEC-1 using urea-denatured acrylamide gel (Fig 7g and 7h). The results confirmed that no specific fragments corresponding to the cleavage of tRNAs were produced. Next, northern blot analysis using probes specific to the 5' or 3' sequence of 28S or 18S rRNA was performed to determine rRNA degradation from HMEC-1. In agreement with prior analysis, an obvious unspecific degradation of rRNAs could be detected (S8 b and c Fig). Collectively, these findings suggested that during *Rh*-B8 infection, both VapC toxins likely gain access to host cell cytoplasm, where they induce host cell arrest through ribosomal RNA damage. Notably, our data indicate that VapC2 appears to play a more prominent role in this process compared to VapC1.

Our above findings regarding VapCs' host-associated action raise a question about their role in modulating host ROS homeostasis, which we previously attributed to host antioxidant response (Fig 3h, 3i and 3j). We then assessed the ROS profile in HMEC-1 cell lines during early-stage infection by *vapC* mutant strains. In HMEC-1 cells infected with Δ*vapC1* mutant strain, ROS levels exhibited a transient increase followed by a rapid decline, mirroring the kinetics observed with wild-type (WT) infection (Fig 7i). Both infections induced comparable upregulation of host antioxidant enzymes (Figs 3h and 7j). We have demonstrated that Δ*vapC1* proliferates uncontrollably upon host cell entry, increasing susceptibility to ROS-mediated damage and exhibited attenuated growth. Despite these growth differences, WT and Δ*vapC1* infections generated nearly identical ROS profiles. This suggests that host ROS dynamics are largely independent of VapC1's activity. In contrast, Δ*vapC2* infection led to accelerated ROS clearance, with levels decline more rapidly compared to WT (Fig 7i). This observation implies that VapC2-mediated translational suppression may impair host antioxidant defenses. Supporting this hypothesis, infection with *Rh*-B8^vapC2+ resulted in longer-lasting ROS accumulation (Fig 7i), accompanied by accelerated host rRNA degradation (S8d Fig). Collectively, these findings suggest that VapC2 may modulate host ROS dynamics via its broad translational suppression activity, while the host antioxidant response appears to be the primary determinant of redox equilibrium during *Rh*-B8 infection.

## Discussion

Eukaryotic hosts have evolved multiple mechanisms to combat invading microbial pathogens. One of the most widely used mechanisms developed by vascular endothelial cells—the primary target cells of SFG *Rickettsia*—is to create an oxidative stress environment by producing ROS [44]. In this study, we found that during the early stages of *Rh*-B8 infection, HMEC-1 had a substantial accumulation of ROS. Since a cell that dies rapidly during infection does not aid intracellular

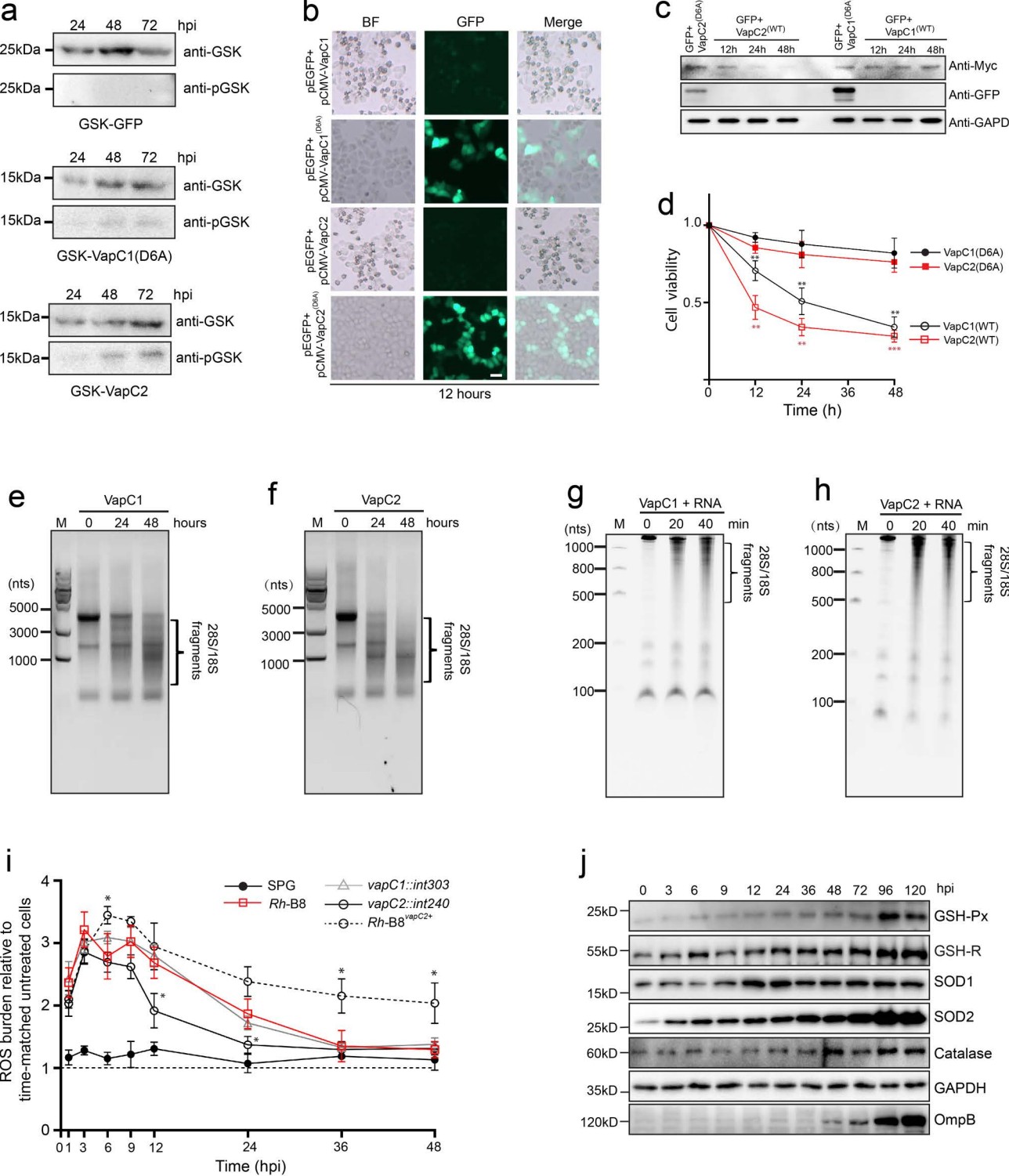

**Fig 7. *Rh*-B8 VapCs degrades rRNAs from human cells.** (a) GSK-tagged inactive VapC1 mutant and wild-type VapC2 and GSK-tagged GFP were expressed in *Rh*-B8. Samples were collected at 24, 48, and 72 hpi (MOI = 5.0). The samples were immunoblotted to detect total GSK-tagged protein with an anti-GSK epitope tag antibody and the GSK-tagged protein upon exposure to host cytosolic kinases with a specific anti-phospho-GSK antibody. GFP-GSK was used as a non-secreted protein control and did not show reactivity with the anti-phospho-GSK antibody. (b) The effect of expression of

wild-type VapCs and the inactive mutants in HEK293 cell lines after transformation for 12 hours. The bright field (BF) images show cytopathic effects upon VapCs' expression. Co-expressed GFP protein was used as a marker for exogenous gene expression. Scale bar, 10 μm. (c) Western blot to show the protein expression of VapC and GFP as described in (b). After co-expression of GFP and inactive VapC mutants for 12 hours, the cell lysates were used as a control. (d) Cell viability upon expression of wild-type VapCs and the inactive mutants. Data represent the mean ± SD of three independent biological replicates (** $p < 0.01$, *** $p < 0.001$ vs. time-matched VapC mutants; two-tailed unpaired t-test). (e) and (f) Agarose gel analysis of total RNA extracted from HEK293 cells upon expression of VapC1 and VapC2. (g) and (h) Total RNA isolated from HMEC-1 was incubated with purified VapCs showing rRNA degradation. The mixture was subjected to a 4.5% denaturing polyacrylamide gel and visualized by ethidium bromide staining. The rRNA degradation could be visualized, while 28S rRNA cannot be separated on the gel due to the large molecular size. (i) Reactive oxygen species (ROS) levels in HMEC-1 cells infected with WT and mutant Rh-B8 strains. ROS abundance was quantified and normalized to time-matched untreated controls. Data represent the mean ± SD of three independent biological replicates (* $p < 0.1$, ** $p < 0.01$, *** $p < 0.001$ vs. time-matched WT strain; two-tailed unpaired t-test). (j) Protein levels of host antioxidant enzymes in HMEC-1 cells following *vapC1::int303* strain infection. Expression of glutathione peroxidase (GSH-Px), glutathione reductase (GSR), superoxide dismutase (SOD1/2), and catalase were assessed at indicated time points post-infection. Bacterial growth was monitored via *Rh*-B8 OmpB levels.

pathogen's proliferation, it was proposed that host-cell death during a rickettsial infection must be tightly controlled to ensure and enhance host cell survival [45,46]. In agreement with this opinion, it was found that the excessive free radicals upon *Rh*-B8 infection could be subsequently neutralized by ECs' anti-oxidative response. This may prevent the generation of reactive oxygen species and the antioxidant defense system's scavenging ability from becoming out of balance, which could ultimately lead to irreversible oxidative stress and quick host-cell death. However, after invading host cells, rickettsiae have to overcome oxidative attacks over a period before excessive host ROS are eliminated.

Prokaryotes may employ TA modules to induce persistence, enabling them to survive in stressful environments. Because of their potential damage to host cells, TA systems have been postulated as incompatible with obligatory intracellular microorganisms [24]. However, they are present and relatively abundant in SFG *Rickettsia*. Moreover, we found that two *vapBC* modules from *Rh*-B8 exhibited a statistically significant increase in transcript levels during the initial stage of infection. Previous research on *M. tuberculosis* revealed that a variety of environmental stresses may cause *vapBC* modules to be activated [47]. Here, we found that rickettsial *vapBC* transcriptions are directly stimulated by *in vitro* $H_2O_2$ exposure, but not other stimuli. Longer antibiotic exposure was also discovered to increase their transcriptions; however, the mechanism is to be further studied, even though the bactericidal activity of antibiotics may also be connected to the induction of ROS [8]. Moreover, the overexpression of Lon protease in *Rh*-B8, which has been shown to promote wide adaptive responses to various survival challenges, was correlated with fluctuations in the transcript levels of *vapBC* modules [48]. An *in vitro* degradation assay was then used to confirm its protease activity against the C-terminal tail of VapB antitoxins. These results indicated that the disintegration of VapBC complex by Lon protease, which was elevated by a stressful oxidative environment, resulted in the release of free toxins and the activation of *vapBC* operons (Fig 8). Another caseinolytic or caspase-3-like protease (Clp) X/P that can alter the proteome and promote broad adaptive responses was also found in the *Rh*-B8 genome. Nevertheless, it was discovered that this protease was inactive during the infection. Furthermore, the hydrophobic residues at the C-terminus of VapBs match the preference of Lon protease for exposed hydrophobic patches [49].

Our results demonstrate that VapC1 does not directly facilitate host ROS clearance following oxidative burst. However, VapC1-induced bacterial dormancy may create a false impression of infection resolution while potentially suppressing host ROS production. We provide evidence that cytoplasmic VapC2 modulates host ROS clearance via translational inhibition, with its biological impact being tightly regulated by the transient expression pattern of the *vapBC* operon. This controlled toxin release mechanism may facilitate precise host manipulation - sufficient to compromise antimicrobial defenses (including ROS scavenging) and optimize the intracellular niche, yet carefully balanced to avoid excessive host cell damage that would threaten bacterial survival. The observed host antioxidant response likely represents a protective mechanism against severe oxidative damage. This interpretation gains further support from our demonstration that $H_2O_2$ treatment during later infection stages (48–72 hpi) no longer activates *vapBC* operon

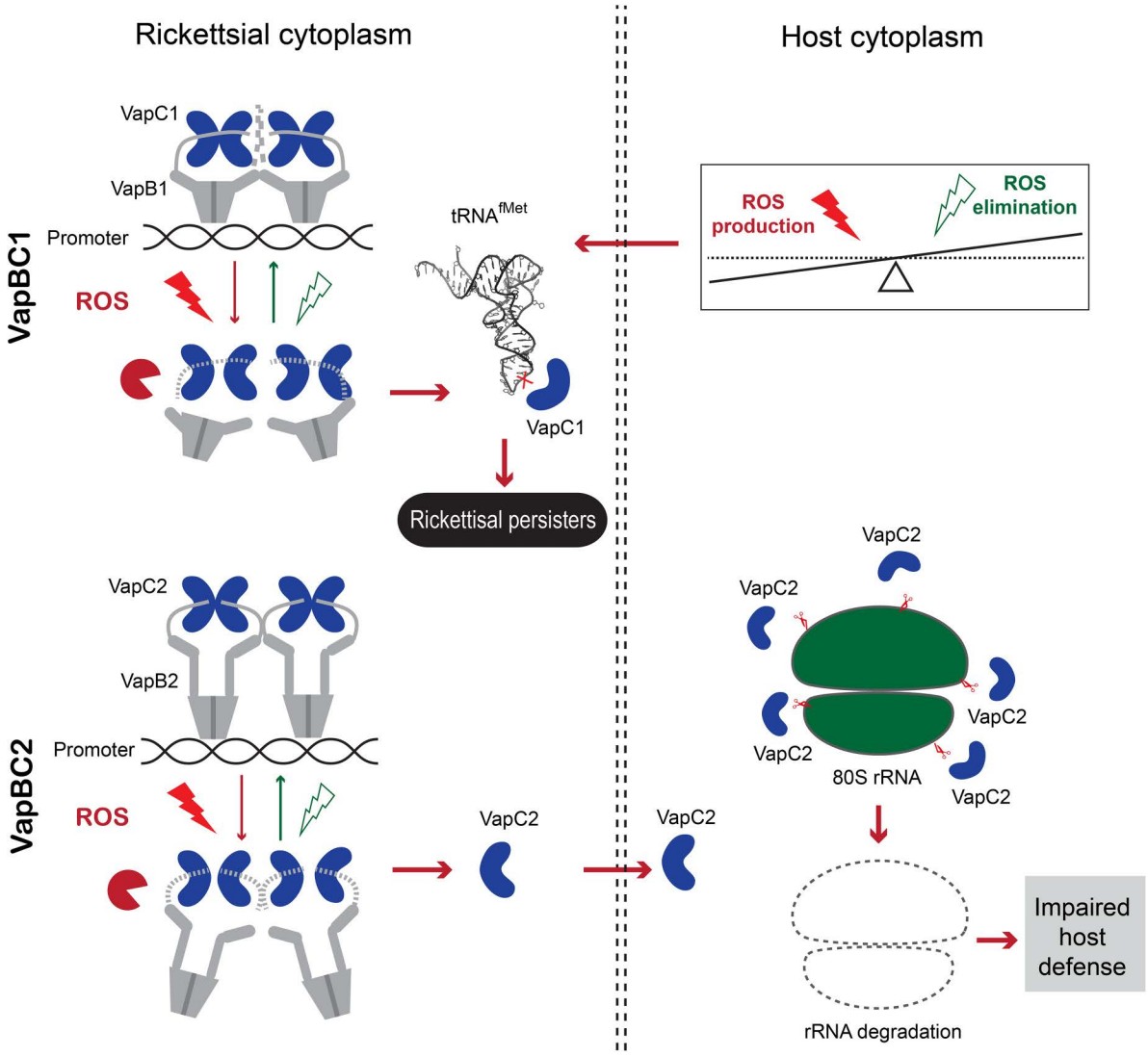

**Fig 8. Proposed model for the stress response regulation by *vapBC* modules.** Under host oxidative crisis, VapC1, which is upregulated in the expression during the early rickettsial infection via degradation of VapB1 antitoxins by Lon protease, cleaves rickettsial initiator tRNA$^{fMet}$, thereby inducing persisters. VapC2 could be exposed to host cytoplasm and impair host anti-infection response by degrading host rRNAs. See also the discussion for details.

transcription, suggesting either establishment of effective host antioxidant defenses, or pathogen metabolic adaptation to the intracellular environment.

Previous research on TA modules from *M. tuberculosis* showed that some of the TA modules were individually dispensable for stress adaptation [47]. Here we found that disruption of *vapC*1 in *Rh*-B8 resulted in a significantly smaller plaque on the HMEC-1 monolayer and the strain is more susceptible to ROS exposure *in vitro*. Furthermore, the Δ*vapC*1 strain was defective in establishing infection in mice. However, the *vapC*2 disruption in *Rh*-B8 produces somewhat smaller plaques but does not affect bacterial growth in mice. These findings suggested that *vapBC*1 module is critical in sustaining bacterial proliferation both *in vitro* and *in vivo*, whereas *vapBC*2 module might not be essential for *Rickettsia*'s multiplication. Overexpression of wild-type VapC1 in *E. coli* results in a significant growth arrest caused

by translation inhibition. Despite several attempts, we were unable to generate a *Rickettsia* strain that expressed wild-type VapC1. In support of this notion, our study revealed that initiator tRNA$^{fMet}$ of *Rh*-B8 is the target of VapC1. The tRNA$^{fMet}$, as a VapC target, was also identified in other bacteria, such as *M. tuberculosis*, *Shigella flexneri*, *Salmonella typhimurium*, and *Haemophilus influenzae* [22,34,50,51]. Since eliminating tRNA$^{fMet}$ should be the most effective way to inhibit protein translation compared to cleaving other tRNAs, the tRNA$^{fMet}$ cleavage might be a universal action of bacterial VapC toxins [22]. These results indicated the ability of free VapC1 toxin to induce rickettsial dormancy, thereby enabling *Rickettsia* to survive oxidative damage after entering host cells. Furthermore, since *Rickettsia* has to deal with various innate and adaptive host immune responses after entering host cells, staying dormant must be an optimal strategy to avoid too extensive host attacks in the early stages of infection, which is also in agreement with the long log phase observed for rickettsial growth.

In contrast to VapC1, VapC2 overexpression showed no significant growth inhibitory effect on *Rh*-B8. In support of this result, no bacterial RNA degradation was detected following VapC2 expression in both *E. coli* and *Rh*-B8. Likewise, inactive VapC toxins were also discovered in *M. tuberculosis* [47]. This raised a possibility that not all VapC toxin was involved in the action within the bacteria. It was proposed that TA toxins might be involved in regulating surrounding bacterial population or damaging eukaryotic cells, while it is unknown whether these toxins can penetrate cell membranes [16,17,19]. However, the proteins released by obligatory intracellular bacteria, unlike those from free-living bacteria, can enter the host cytoplasm directly. Here, we discovered that both *Rh*-B8 VapC toxins were capable of entering host cytoplasm and producing adverse effects on human cells during infection; however, the VapC2 toxins induced more serious damage. This result is consistent with the finding that, as compared to the WT strain, the *Rh*-B8 strain expressing VapC2 produces larger plaques and causes more rapid host cell death. It should be noted that online prediction for secreted signal peptides of both VapC toxins did not yield any reliable results. Therefore, there is the possibility that VapCs were secreted as a folded protein by *Rickettsia* secretion pathways, such as the twin-arginine translocation (Tat) system [52], and the possibility that VapC toxins were exposed to host cytoplasm after bacterial lysis cannot be ruled out. In support of above notion, we discovered that both VapC toxins non-specifically degraded ribosomal RNAs from human cells. Unlike in *Rickettsia*, in which VapC toxicity would be neutralized by the cognate VapB antitoxin once the oxidative stress crisis has been solved, the toxins exposed in the host cytoplasm may result in sustained host protein translation disruption (Fig 8). As previously mentioned, many bacterial TA modules were discovered to be unrelated to functions within the bacteria. For instance, *M. tuberculosis* has more than 80 TA modules, 48 of which are *vapBC*, and numerous toxins have been demonstrated to have no effect on the bacteria itself [22]. Therefore, systematic investigation of host molecular targets for these toxin-antitoxin modules is warranted to establish their true physiological functions during infection.

In summary, our *in vitro* and *in vivo* data demonstrated that both *vapBC* TA modules contribute to *Rh*-B8's intracellular survival and pathogenesis. We discovered that TA modules are compatible with obligate intracellular bacteria *Rickettsia* and induce persisters to overcome host ROS damage, which helps infection establish during the early stage. Notably, VapC toxins may reach host cytosol and target host rRNAs during infection, which undermine host anti-infection defenses and produce a favorable host environment. This study highlights the adaptive strategies employed by SFG rickettsia to thrive in the hostile intracellular environment and provides a novel insight into the interplay between host-associated pathogens and their hosts.

## Materials and methods

### Ethics statement

All animal experiments were performed in accordance with institutional ethical guidelines and approved protocols from the Beijing Institute of Microbiology and Epidemiology (IACUC-IME-2023–001) and Anhui Medical University's Ethics Committee (No. LLSC20200351).

## Bacterial strains, growth, and purification

*R. heilongjiangensis* strain B8 was isolated and preserved by our laboratory [4]. *Rickettsia* was cultivated and propagated at 34 °C in Vero cells (African green monkey kidney epithelial cells) in DMEM plus 2% FBS (fetal bovine serum) or HMEC-1 (human microvascular endothelial cells) in HMEC media (ECM basic media supplemented with 5% FBS, 10 mM L-glutamine, 10 ng/mL epidermal growth factor, 1 μg/mL hydrocortisone) at a multiplicity of infection (MOI) of 0.1 unless otherwise specified. For the purification, infected cells together with the medium were harvested and centrifuged at 12,000×g for 5 min at 4 °C. The pellets were suspended in cold SPG (sucrose-phosphate-glutamic acid–Mg) buffer (220 mM sucrose, 3.8 mM $KH_2PO_4$, 8 mM $K_2HPO_4$, 5 mM potassium glutamate, and 10 mM $MgCl_2$) and transferred to a precooled 15 mL conical centrifuge tube containing 1 mm glass beads. Cells were lysed by three rounds of vigorous vortex (15 s) with 30 s intervals of incubation on ice. The cellular debris and unbroken cells were removed by centrifugation at 1,200×g. The bacteria in the supernatant were collected by centrifugation at 10,000×g for 15 min at 4 °C. The rickettsial pellet was resuspended in SPG buffer and directly used for the following plaque assay, electroporation, or quantitative analysis. For strain preservation, the supernatant was divided into 1.5 mL tubes with roughly $1 \times 10^{10}$ bacterial cells per mL after being rinsed three times with cold 250 mM sucrose and frozen at -80 °C.

## Plasmids construction

All restriction enzymes, ligases, and DNA polymerases used in this study were purchased from New England BioLabs (NEB) unless otherwise specified. Primers, oligonucleotides, and genes were ordered from General Biol Co. and the sequences were listed in S2 Table. All cloning was carried out in *E. coli* DH5 competent cells (Takara). For the construction of plasmids for wild-type VapC toxin expression and purification in *E. coli* (pETDuet-N-VapBC1$^{WT}$, pET28a-VapB1$^{WT}$, pET28a-VapB2$^{WT}$, and pET28a-VapC2$^{WT}$), optimized gene sequence coding VapB1 and VapC1 for expression in *E. coli* K-12 was synthesized (S2 Table), followed by subcloned into MCS-1 of pETDuet-N using BamHI/HindIII and MCS-2 of pETDuet-N using KpnI/XhoI, respectively. VapBC1 complex was expressed upon induction of IPTG (isopropyl-D-1-thiogalactopyranoside), which was then used for the purification of wild-type VapC1. The Optimized genes coding for VapB1, VapB2, and VapC2 were ligated into a pET28a vector using BamHI and XhoI (S2 Table). N-terminal 6×his tagged proteins were expressed upon induction of IPTG. For the construction of plasmids to express GST-tagged proteins (pGEX-6p-1-VapB and pGEX-6p-1-La), DNA coding the wild-type VapB and truncated mutants were amplified from *Rh*-B8 chromosome DNA using corresponding primers as shown in S2 Table. PCR products were digested with EcoRI and XhoI, and ligated into vector pGEX-6p-1. To construct pGEX-VapB1-41–53, pGEX-VapB2-82C, pGEX-VapB2-66–81, and pGEX-VapB2-42–66, full-length DNA oligonucleotides with cohesive ends were ordered, annealed and ligated to EcoRI/XhoI digested pGEX-6p-1 directly. Lon (ATP-dependent protease La) coding gene was amplified from *Rh*-B8 chromosome DNA using primers Fw_La and Rv_la (S2 Table), followed by digestion and ligation to pGEX-6p-1 as described above. This plasmid includes a Tev protease site that can be utilized to remove the GST tag during the subsequent purification process. To construct the plasmids for protein expression induced by arabinose in *E. coli* (pBAD-VapC1$^{WT}$, pBAD-VapC2$^{WT}$, pBAD-VapC1$^{D6A}$, pBAD-VapC2$^{D6A,}$ and pBAD-VapC20-Mtb), VapC1$^{WT}$ and VapC2$^{WT}$ coding sequences were amplified from the synthetic DNA template codon optimized for *E. coli* K-12 using primers Fw_VapC1-wt-X and Rv_VapC1-wt-E, Fw_VapC2-wt-X and Rv_VapC2-wt-E, respectively (S2 Table). PCR products were digested with XhoI and EcoRI and ligated into vector pBAD33. The VapC1$^{D6A}$ and VapC2$^{D6A}$ was amplified from pBAD-VapC1$^{WT}$ and pBAD-VapC2$^{WT}$ with primers Fw_VapC1-D6A-X and Rv_VapC1-wt-E, Fw_VapC2-D6A-X and Rv_VapC2-wt-E, respectively (S2 Table). The PCR products were digested with XhoI and EcoRI and ligated into pBAD33. The synthetic *M. tuberculosis* VapC20 coding gene was digested and ligated to pBAD33 as described above (S2 Table). The plasmids for expression of VapC in *Rickettsia* (pRAMF2-VapC1$^{WT}$, pRAMF2-VapC1$^{D6A,}$ and pRAMF2-VapC2$^{WT}$). The rickettsial shuttle vector pRAMF2 that allowed for recombinant expression of N-terminal FLAG-tagged proteins in *Rickettsia* driven by the *ompB* promoter was kind provided by T. Hackstadt [40]. The wild-type VapC1 and VapC2 genes were amplified from

*Rh*-B8 chromosome DNA using primers Fw_VapC1-wt-BW and Rv_VapC1-wt-BH, Fw_VapC2-wt-BW and Rv_VapC2-wt-BH, respectively. The PCR products were digested with BsiWI and BssHII and ligated to pRAFM2. To construct pRAMF2-VapC1$^{D6A}$, the coding gene was amplified from pRAMF2-VapC1$^{WT}$ with primers Fw_VapC1-D6A-BW and Rv_VapC1-wt-BH. The PCR products were then digested with BsiWI and BssHII and ligated to pRAFM2. Plasmids for expression of the protein in HEK293 cells (pCMV-N-Myc-VapC1$^{WT}$, pCMV-N-Myc-VapC1$^{D6A}$, pCMV-N-Myc-VapC2$^{WT}$, pCMV-N-Myc-VapC2$^{D6A}$). The VapC1 and VapC2 coding genes were optimized for expression in human cell lines and synthesized (S2 Table), followed by digesting with BamHI and XhoI and ligated to pCMV-N-Myc (Beyotime). To construct the mutant expression vector, VapC1$^{D6A}$, and VapC2$^{D6A}$ coding genes were amplified from pCMV-N-Myc-VapC1$^{WT}$ and pCMV-N-Myc-VapC2$^{WT}$ with primers Fw_VapC1-optfh-D6A-B and Rv_VapC1-optfh-X, Fw_VapC2-optfh-D6A-B and Rv_VapC2-optfh-X, respectively. The PCR products were digested with BamHI and XhoI and ligated to pCMV-N-Myc. Plasmids for gene disruption in *Rickettsia* (pARR-vapC1-304s and pARR-vapC2-241s). The targetron vector pARR modified based on pACDK4 was kindly provided by T. Hackstadt [38]. Appropriate intron insertion sites into *vapC* of *Rh*-B8 were determined through the TargeTron software algorithm (www.sigma-aldrich.com/targetronaccess), which predicts high-specificity group II insertion locations in DNA sequences. The intron target positions for *vapC1* and *vapC2* were selected at 304 and 241 bp downstream of the start codon, respectively. The corresponding EBS2, EBS1d, and IBS primers were assembled and ligated to pARR following TargeTron's protocol (S2 Table). All the constructions shown above were confirmed by DNA sequencing.

## Growth curves

An OmpB coding gene fragment (B8-OmpB-frag-RTP as shown in S2 Table) was synthesized and inserted into the pUC57 vector, resulting in pUC57-OmpB-frag. The standard curve for measuring the copy number of the *ompB* gene was first created using plasmid pUC57-OmpB-frag as the standard template, using primers Fw-RTP-B8-OmpB, Rv-RTP-B8-OmpB, and Probe-B8-OmpB (S2 Table). The copy numbers of *Rh*-B8 could be determined using the output formula $Y = 3.494X + 44.716$, where X is the bacterial number expressed as a Log10 value and Y is the cq value. To establish the growth curves of *Rh*-B8 in HMEC-1, the amount of live rickettsia was first determined using the plaque assay on Vero cell monolayers as stated below. Growth curves were then carried out following infection of HMEC-1 at MOI of 0.1 in 6-well plates. To collect all bacterial cells, the cells and medium were centrifuged at $12,000 \times g$ at each time point. For the growth curves calculation using qPCR, bacterial genome DNA was extracted using MiniBEST Universal Genomic DNA Extraction Kit (Takara) following manufacturers' instructions. The extracted genomic DNA was then used as a template for quantitative analysis, using the primers and probes as described above. To determine the growth curves using the plaque assay, the collected bacteria were serially diluted and then inoculated onto Vero or HMEC-1 cell monolayers according to the procedures for the plaque assay described below.

## Host cell viability assay

HMEC-1 cell lines inoculated with *Rh*-B8 at MOI of 0.1 were incubated at 34 °C, and samples were collected at indicated time points. HEK293 cell lines transformed with plasmids expressing VapCs (Lipofectamine 2000, Invitrogen) were collected at indicated time points. Cell samples were incubated with test compounds at the indicated concentrations and a cell viability assay was performed following the manufacturers' suggestion (CCK-8, Beyotime)

## *In vitro* susceptibility assay

To test the transcription level of *vapBC* modules upon exposure to various stress conditions, HMEC-1 cell lines were first inoculated with *Rh*-B8 at a MOI of 0.1 for 12 hours, and then the medium was replaced with HMEC medium supplemented with 10 μM $H_2O_2$, 25 μM $NaNO_2$, zero FBS, 2.5 μg/mL of chloramphenicol, or 2.0 μg/mL of rifampin, respectively. The samples at 3, 6, 9, 12, and 24 hours after stimulation were collected and used for transcription analysis as described

below. To measure the susceptibility of wild-type *Rh*-B8 and mutants to oxidative stress, the cells were first inoculated with each strain at a MOI of 5.0 and then cultured with 100 μM $H_2O_2$ for 24 hours. For bacterial enumeration, all bacteria were collected by centrifuging at 12,000 × g, followed by a 10-fold serial dilution and plaque assay.

## qRT-PCR

To determine the transcription level of *Rh*-B8 genes during the infection or after various stress stimulation, RNAiso Plus (Takara) was used to extract total RNA at each post-infection time point, followed by reverse transcription (RT) utilizing PrimeScript RT reagent kit with gDNA Eraser (Perfect Real Time) (Takara). The relative quantitative PCR was carried out using TB Green Premix Ex Taq kit (Takara), following the manufacturer's instructions. The primers used in this work are listed in S2 Table. In this study, the targeted mRNAs were normalized to the *ompB* (*Rh*-B8) or GAPDH (Human cells) mRNA expression. The $2^{-\Delta\Delta Ct}$ algorithm was used for the calculation of fold change.

## Electroporation

Purified *Rh*-B8 was transformed with plasmids as previously described [53]. In brief, approximately $10^9$ bacterial cells isolated from Vero cell monolayers in one T75 flask were suspended in 50 μL of 250 mM sucrose and used for one electroporation experiment. Following mixing with 20 μg of plasmids, the suspension was pipetted into a 0.1-cm gap electroporation cuvette (BTX Electronic Genetics) and allowed to chill for ten minutes on ice. The cuvette was electroporated (high voltage = 1750V, pulse time = 5 ms, resistance = 150 Ω, capacitance = 36 μF) in a BTX Electro Cell Manipulator (ECM 630). Following electroporation, the suspension was immediately diluted into 500 μL of Vero cell medium and added to six-well plate Vero cell monolayers. The cells were incubated for 30 minutes at 37 °C, and then 2.5 mL of cell media was added. The cells were then incubated for overnight at 34 °C. Rifampin was used in the transformant clone selection process as described below.

## Plaque assays and clonal isolation

Vero or HMEC-1 cell monolayers on six-well plates were inoculated with 500 μL of diluted bacteria ($10^{-3}$-$10^{-8}$ in the appropriate cell media). To allow for optimal adhesion and infection, the cells were incubated at 37 °C for 30 minutes. Afterward, cell medium was added, and the cells were incubated at 34 °C for overnight. After that, the media was removed and then replaced with the cell medium that contained 0.8% agarose and 5% fetal bovine serum (FBS). In this stage, rifampin was added to the culture in a final concentration of 2.0 μg/mL to select transformed bacteria. The plates were incubated at 34 °C until plaques formed, which could normally be visualized after approximately 7 days. To image plaques, neutral red (0.01% final concentration) in cell medium with 2% FBS and 0.8% agarose was applied over the prior agarose layer. Plaques were counted and photographed 24 hours following the addition of neutral red. To select clonal transformants, plaques were picked, recloned, and expanded by inoculation on fresh Vero cell monolayers in the presence of 2.0 μg/mL rifampin three times. To confirm the transformants overexpressing VapC or mutants, the rifampin-resistant plaques were first confirmed to express a green fluorescent protein (GFP) by fluorescence microscopy after expansion on Vero cell monolayers (S5c Fig). The presence of the *gfp* and *arr* genes was confirmed by PCR with the primers GFP-F/GFP-R and arr-F/arr-R, respectively (S2 Table). For verification of transposon insertion in the VapC coding gene, plaques after rifampin-resistant selection were inoculated and expanded in Vero cell monolayers. Total DNA was then extracted and used as the template in PCR, with primers Fw_VapC1-wt-X and Rv_VapC1-wt-E or Fw_VapC2-wt-X and Rv_VapC2-wt-E (S2 Table). PCR products were analyzed on a 1% agarose gel, and DNA fragments with apparent larger MW were sent for sequencing directly (S3 Fig).

## Protein expression and purification

*E. coli* BL21 (Takara) transformed with expression vectors pETDuet-N-VapBC1WT, pET28a-VapB1WT, pET28a-VapB2WT, pET28a-VapC2WT, and pGEX-6p-1-La were induced with 0.5 mM IPTG overnight at 18 °C under vigorous shaking.

Bacterial cells were lysed by sonication and the raw extract was cleared by centrifugation at 12,000 × g for 30 min. For purification of wild-type VapB1, VapB2, and VapC2, the lysate was incubated with 200 μL of Ni-NTA resin (Takara) overnight at 4 °C. The resin was then loaded onto a gravity-flow column and washed with 10 column volumes of wash buffer (50 mM Tris-Cl, pH 7.4, 300 mM NaCl, 10 mM imidazole, 5% glycerol, and 5 mM DTT). Proteins were eluted using elution buffer (50 mM Tris-Cl, pH 7.4, 300 mM NaCl, 150 mM imidazole, 5% glycerol, 5 mM DTT). Fractions containing targeted proteins were pooled and dialyzed overnight against 50 mM Tris-Cl, pH 7.4, 150 mM NaCl, 5 mM $MgCl_2$, and 1 mM DTT, following confirmation of protein expression using SDS-PAGE analysis. To purify wild-type VapC1, lysates containing VapBC1 complex together with Ni-NTA resin were loaded onto a gravity-flow column as described above. The complex was washed using denaturing buffer (50 mM Tris-Cl, pH 7.4, 300 mM NaCl, 7 M urea, and 1 mM DTT) to remove VapB1, followed by on-column renaturation using 50 mM Tris-Cl, pH 7.4, 150 mM NaCl and 1 mM DTT. The His-tagged VapC1 was finally eluted using the elution buffer as described above. After confirmation with SDS-PAGE, the elutes were dialyzed overnight against 50 mM Tris-Cl, pH 7.4, 150 mM NaCl, 5 mM $MgCl_2$ and 1 mM DTT at 4 °C. For the purification of Lon protease, the lysate was mixed with 200 μL of glutathione-superflow resin (Takara) and washed with phosphate-buffered saline (PBS) buffer (pH 7.4) following the batch purification procedure (Takara). GST-tagged TEV protease (Beyotime) was added into the resin and incubated at 4 °C overnight with gentle rotation. After centrifugation at 800 rpm, the supernatant containing GST-tag-free Lon protease was collected and subjected to SDS-PAGE analysis. Purified proteins were up-concentrated to ~ 4 mg/mL using 15 mL Amicon ultra centrifugal filters (Millipore, 10K for VapC, 50K for Lon protease) and stored at -80 °C.

## GST-pull down assay

The GST-VapB fusions were induced and expressed in *E. coli* BL21. Glutathione resin (50 μL) was added to cell lysates, and the mixture was then rinsed three times with 50 mM Tris-Cl, pH 7.4, and 150 mM NaCl. After equilibrating with purified VapC1 or VapC2 (200 μg) in 100 μL of buffer (50 mM Tris-Cl, pH 7.4, 150 mM NaCl), the immobilized GST fusion proteins were incubated for one hour at 4 °C. Following three rounds of washing in the same buffer, the bound protein-containing resin was separated using 15% SDS-PAGE and visualized using Coomassie blue staining.

## ROS and GSH measurement

HMEC-1 cells were inoculated with *Rh*-B8 at 0.1 MOI and cultured as previously described. To measure ROS, cells at selected post-infection time points were treated with 10 μM 2',7'-Dichlorodihydrofluorescein (DCFH-DA) (Beyotime) according to the manufacturer's instructions. The fluorescence intensity of 2',7'-dichlorofluorescein (DCF) was determined using flow cytometry. The data was presented as a fold change for each sample compared to HMEC-1 at each time point without infection. To measure the amount of reduced GSH in HMEC-1 upon infection by *Rh*-B8, the cell lysates at each post-infection time were incubated with 0.5 mM DTNB and detected at the absorption of 412 nm following the protocol of GSH and GSSG assay kit (Beyotime).

## *In vitro* cleavage assay

To determine the proteolytic activity of *Rh*-B8 Lon protease against VapB antitoxins, cell lysates containing GST-tagged VapB were subjected to glutathione resin (Takara) and washed with buffer containing 50 mM Tris-HCl, pH 7.4, 10 mM $MgCl_2$, 1 mM DTT, and 5 mM ATP. The resin was incubated with 1 μg of Lon and gently rotated for 15–30 minutes at 37 °C. The reaction was terminated by adding 2 × Protein SDS-PAGE loading buffer (Takara) and heating for 5 minutes at 100 °C. The cleavage results were assessed via 12% SDS-PAGE and Coomassie blue staining.

For *in vitro* RNA cleavage assay, the phenol/chloroform method was used to prepare the total RNA of *E. coli*, *Rickettsia*, and HMEC-1 cells (RNAiso plus, Takara) or small RNA of *Rh*-B8 (RNAiso for small RNA, Takara) following the protocol's

instruction. VapC solution was first exchanged into cleavage buffer (20 mM HEPES, pH 7.5, 50 mM KCl, 10 mM MgCl$_2$, 20% glycerol, and 2 mM DTT) through a desalting column (GE Healthcare). Purified total RNA (10 µg) was then incubated with VapC (2 µg) at 37 °C in a volume ratio of 1:1. Following phenol/chloroform extractions to inactivate the VapC protein, 2.5 times the volume of ice-cold ethanol and 0.3 M Na-acetate were used to precipitate the cleaved RNA for 30 minutes at -80 °C. The RNA pellets were resuspended in nuclease-free water and subjected to either agarose gel or urea-denature acrylamide gel.

## Metabolic labeling

*E. coli* BL21 harboring the pBAD empty vector, pBAD-VapC1$^{WT}$ or pBAD-VapC2$^{WT}$ were grown exponentially in M9 medium supplemented with 0.2% glucose and all amino acids except methionine at 37 °C until OD$_{600}$ ~0.2. Toxin expression was induced with 0.2% L-arabinose at time zero. At each time point, 1 mL culture was removed for OD measurement, and another 1 mL sample was collected and pulsed for 1 min with radiolabeled $^{35}$S-methionine (10 µCi) or $^3$H-uridine (5 µCi), at 37 °C. Subsequently, protein and nucleic acids were precipitated by adding ice-cold trichloroacetic acid (TCA) to a final concentration of 20%, followed by centrifugation for 15 min at 4 °C. The final pellet was washed twice with ice-cold 100% ethanol and resuspended in 200 µL of 1% SDS. The incorporation of radioisotopes was measured using a Beckman LS 6000SC liquid scintillation counter. The amount of radioisotope incorporated was normalized to the OD$_{600}$ at each time point.

## CRISPR/Cas9 knockout

The *Keap1*-knockout HMEC-1 cell line with impaired ROS production was generated using CRISPR/Cas9 gene editing. Briefly, single-guide RNAs (Keap-KO-1 and Keap-KO-2 in S2 Table) targeting *Keap1* (Gene ID: 9817) were designed using CHOPCHOP online server and cloned into the lentiCRISPRv2 plasmid (Addgene 52961). Lentivirus was produced in HEK293T cells co-transfected with the sgRNA constructs, psPAX2 (Addgene 12260), and pMD2.G (Addgene 12259). For *Keap1* knockout (KO) generation, HMEC-1 cells were transduced with lentivirus containing two sgRNAs simultaneously, selected with 2 µg/mL puromycin for 48 hours, and then cultured without puromycin for 7 days. Successful knockout was confirmed by Western blot analysis using anti-Keap1 antibody (mAb 8047, CST), and the validated KO pool was expanded for subsequent *Rh*-B8 infection experiments.

## Western blot analysis

To detect anti-oxidant enzymes in HMEC-1 after *Rh*-B8 infection, infected cells at each time point were boiled in 1x SDS loading buffer (Takara), separated by 12% SDS-PAGE, and transferred to a PVDF membrane (Millipore, IPFL00010). The membrane was blocked overnight at 4 °C in TBST buffer (20 mM Tris-Cl, pH 8.0, 150 mM NaCl, and 0.1% Tween 20) with 5% dry milk (Beyotime). Primary antibodies anti-SOD2 (mAb 13141, CST), anti-SOD1 (mAb 37385, CST), anti-GPX1 (mAb 3286, CST), anti-catalase (mAb12980, CST), and anti-GSR (mAb 62448, CST) were diluted in TBST buffer following the instructor's recommendation and incubated with the membrane at 4 °C overnight. Blots were washed in TBST buffer and treated with the appropriate secondary antibody. Blots were processed and scanned using the Tanon 5200 automatic luminescence imaging equipment (Tanon, Shanghai) according to the manufacturer's instructions. In the GSK secretion assay, secreted GSK-tagged proteins were detected via immunoblotting with a Phospho-GSK-3β (Ser9) antibody (mAb 5558, CST), and the total (non-phosphorylated and phosphorylated) GSK-tagged proteins were detected using the GSK-3β-Tag antibody (9325, CST). To assess protein expression suppression upon VapC expression in mammalian cells, Myc-Tag Antibody (2272, CST) was used to detect expression of VapC toxins, and primary anti-GFP antibody (66002–1-Ig, Proteintech) was applied to detect GFP protein. To identify the multiplication of *Rh*-B8, the anti-OmpB rabbit polyclonal antibody was generated in-house against the purified N-terminal region (amino acids 1–130) of OmpB protein.

## RNA-seq experiments

The method for mapping RNA after overexpression of VapC toxins in *E. coli* follows the protocol as previously published [42], with some modifications. In brief, the RNA fragments were first excised from the urea-denaturing PAGE gel and purified using the crash-soaking method. The sample was ligated to a preadenylated 3'-linker: 5'-rAppCTGTAGGCACCATCAAT-NH2–3' (NEB) with T4 RNA ligase 2, truncated K227Q (NEB), and then treated with T4 Polynucleotide Kinase (NEB) to convert 5'-OH to 5'-P, allowing T4 RNA ligase to ligate 5'-RNA-adpator (S2 Table). Reverse transcription was performed using PrimeScript II High Fidelity RT-PCR Kit (Takara) with primer 3'-RT-Primer-BYT (S2 Table). The mixture was used as the template to amplify DNA fragments with primers: 5'-primer-MS and 3'-RT-Primer-BYT (S2 Table). The PCR products were subjected to a 1% agarose gel and the amplified DNA below 100 bp was isolated after gel excision and sent for secondary generation of DNA sequencing (Biozeron, Shanghai).

## Northern blotting analysis

The total small RNA from *Rh*-B8 was isolated and incubated with pure VapC protein as described above to identify tRNA fragments. The RNA pellets were resuspended in 1 × RNA loading dye (Takara) and heated to 85 °C for 5 minutes before cooling on ice. The samples were separated by electrophoresis using an 8% polyacrylamide-urea gel. RNA was then electroblotted onto a nylon membrane HYBOND N+ (GE Healthcare) in 0.5 × Tris-borate-EDTA (TBE) buffer using a semi-dry transmembrane equipment (Bio-Rad) at 200 mA for 20 minutes. The RNA was cross-linked to the membrane with a UV cross-linker (SCIENTZ03-II) at 1.5 J/cm$^2$ for 90 seconds. The membrane was pre-hybridized at 68 °C for 2 hours in 1 × PerfectHyb Plus hybridization solution (Sigma) before adding digoxin-labeled oligonucleotide probes at a final concentration of 50 ng/mL. The hybridization was carried out for 12 hours at 68 °C. The membrane was then balanced in 2 × SSC, 0.1% SDS for 5 minutes at 68 °C before being washed three times with 0.5 × SSC, 0.1% SDS at the same temperature. After 1 hour of blocking at room temperature, HRP-labeled digoxigenin mAb (Cat No. 49620, Cell Signaling Technology) was incubated with the membrane for 2 hours. Blots were processed and scanned using the Tanon 5200 automatic luminescence imaging equipment (Tanon, Shanghai), following the manufacturer's instructions. For detection of tRNA from *Rh*-B8, the 5'-digoxin labeled DNA probes, including B8-tRNA-fMet-1, B8-tRNA-fMet-2, B8-tRNA-Asp, B8-tRNA-Thr, and B8-tRNA-Gly were used (S2 Table). For the detection of ribosome RNAs, 15 µg of total RNA was separated on a 4.5% denaturing polyacrylamide gel. RNA was transferred and hybridized as described for tRNA. For the detection of 23S rRNA and 16S rRNA from *E. coli*, the following DNA probes were used: EC-16S-5, EC-16S-3, EC-23S-5, and EC-23S-3(S2 Table); For the detection of 28S rRNA and 18S rRNA from HMEC-1, the following DNA probes were used: H-28S-5, H-28S-3, H-18S-5 and H-18S-3 (S2 Table).

## Mouse studies

The animal study was conducted at a BSL-3 facility at the State Key Laboratory of Pathogens and Biosecurity, Academy of Military Medical Sciences. All mice utilized in this investigation were knockouts for the genes encoding IFN-I receptors (Ifnar1) (*Ifnar1*$^{-/-}$) on the C57BL/6J background [39]. 100 µL of wild-type *Rh*-B8 or mutant strain (1.0 × 10$^5$ PFU) in ice-cold PBS was injected into the lateral tail vein using a mouse restrainer. At the time of the initial infection, all mice were between 8 and 12 weeks old and healthy. Each experimental group included mice of both sexes.

## Statistics

The statistical parameters and significance were provided in the figure legends. Statistical analysis and graph production were performed using GraphPad Prism version 8.0.

## Supporting information

**S1 Fig. Transcription level of the TA modules from *Rh*-B8.** (a) Diff quick staining to show the rickettsial multiplication. (b-d) Genomic localization of *yefM-yoeB* and *hicAB* TA modules and their transcription level during bacterial growth. Gene transcription levels were measured after normalization to levels of *ompB*. The data was shown as the fold change in comparison to 0 hpi. Data with the mean ± SD are from n = 3 independent experiments, each with three technical replicates. (TIF)

**S2 Fig. The predicted structure of *Rh*-B8 VapBC modules.** Protein structures were calculated in SWISS-MODEL based on Alphafold DB models. All output models had average model confidence (pLDDT) scores greater than 0.95. (a) and (b) Cartoon diagrams of the VapB and VapC monomers. Secondary structure elements were labeled, and colored in blue (α helices) and grey (β strands), respectively. The residues in the active sites were shown as sticks. The C-terminal toxin interaction domain and N-terminal DNA-binding domain were indicated. (c) The dimeric phd-like DNA binding domain of two VapB antitoxins. (d) Two orthogonal views of DNA-binding model of VapB tetramer. (e) Relative densitometric ratios of VapC1 bands normalized to the control (lane 9, Fig 2d). (f) Relative densitometric ratios of VapC2 bands normalized to the control (lane 14, Fig 2e). Data in (e) and (f) represent mean ± SD from three independent experiments. Statistical significance (*p < 0.1, **p < 0.01, ***p < 0.001, ****p < 0.0001) was determined by unpaired t-test comparing to wild-type. (g) and (h) Top views of the VapB C-terminus interacting with the VapC dimer. The VapC dimer was shown as a surface model, and the VapB C-terminal region was shown as a cartoon model. The C-terminal loop from VapC1 as shown in (e) was long enough to interact with two VapCs simultaneously. (TIF)

**S3 Fig. Transcriptional analysis of stress-induced protease genes in *Rh*-B8.** (a, b) Transcript levels of *Lon* and *ClpX* proteases, normalized to *ompB* expression, presented as fold change relative to 0 hpi. Data represent mean ± SD from three independent experiments, each performed with three technical replicates. (TIF)

**S4 Fig. Identification of *vapC* gene depletion strains and characterization of growth kinetics.** (a,b) PCR amplification of *vapC1* and *vapC2* genes from representative transformants following plaque selection and expansion. The amplified products showed bands approximately 1500 bp larger than wild-type (arrows), confirming correct targeted insertion. These bands were gel-purified and subjected to direct sequencing. (c, d) Growth kinetics and host cell viability during infection with (c) vapC*1::int303*, and (d) *vapC2::int240* strain in HMEC-1 and HMEC-1 (*Keap1-/-*) cells. Data represent mean ± SD from three independent experiments. Statistical significance (*p < 0.1, **p < 0.01, ***p < 0.001) was determined by two-tailed unpaired t-test comparing *Keap1-/-* to wild-type HMEC-1 cells. (TIF)

**S5 Fig. Gross pathology of mice died of *Rh*-B8 infection.** (a) The representative images of spleen tissue from *Ifnar*1-/- mice that died of infection with WT, vapC*1::int303*, and *vapC2::int240* strain via intravenous route were shown. (b) The representative views of immunohistochemical staining on spleen tissue sections. The anti-OmpB antibody was used to indicate the bacteria. Scale bar, 50 µm. (TIF)

**S6 Fig. The expression of VapCs in *E. coli* and *Rh*-B8.** (a) The construction of plasmids for the co-expression of VapB antitoxins and 6 × his tagged VapC toxins in *E. coli*. (b) The pRAMF2 plasmid for expressing wild-type VapCs and mutants in *Rh*-B8. The N-terminal FLAG-tagged target proteins are expressed by an *ompB* promoter, and an *ompA* promoter expresses GFP. (c) Representative images for transformed *Rh*-B8 expressing wild-type VapC2 and GFP after plaque selection. Scale bar 10 µm. (TIF)

**S7 Fig. RNA sequencing to identify potential RNA targets.** (a) Analysis of total RNA from *E. coli* after induction of VapC2. *E. coli* harboring pBAD-VapC2 and pBAD-VapC2$^{D6A}$ were grown in the LB medium, and transcription was induced at time zero by adding L-arabinose (0.2%). VapC20 from *M. tuberculosis* was used as a control. Cell samples were collected at the indicated time points (min). Total RNA extracted from the samples was separated on a 6% denaturing polyacrylamide gel and visualized by ethidium bromide staining. The VapC20 cleavage products were indicated with an arrow. (b) and (c) Northern blot analysis on the total RNA isolated from *E. coli* upon induction of VapC2, using probes specific to the 5' or 3' sequence of 23S rRNA and 16S rRNA of *E. coli*. (d) RNA-seq protocol to identify VapC cleavage products as described in Materials and Methods. (e) Secondary structure diagrams of tRNA$^{Asp}$, tRNA$^{Thr}$, and tRNA$^{Gly}$ identified as potential VapC cleavage products. The arrows indicate the site of cleavage based on sequencing results. (f) Sequence alignment of the potential target tRNAs from *E. coli* and *Rh*-B8. The anticodon sequences were underlined. (TIF)

**S8 Fig. Host rRNA degradation by *Rh*-B8 VapCs.** (a) The effect of expression of wild-type VapCs and the inactive mutants in HEK293 cell lines after transformation for 24 hours. The bright field (BF) images show cytopathic effects upon VapCs' expression. Co-expressed GFP protein was used as a marker for exogenous gene expression. Scale bar, 20 μm. (b) and (c) Total RNA isolated from HMEC-1 was incubated with purified VapCs and then analyzed by Northern blot with probes specific to 5' or 3' sequence of 28S rRNA and 18S rRNA from HMEC-1 cells. (d) rRNA degradation in HMEC-1 cells infected with WT or VapC2-overexpressing strains. Total RNA was isolated from infected HMEC-1 cells, separated on a 4.5% denaturing polyacrylamide gel, and stained with ethidium bromide. Enhanced rRNA degradation was observed in cells infected with VapC2-overexpressing strain compared to WT-infected cells. (TIF)

**S1 Table. Toxin-antitoxin modules in *Rickettsiae*.** (DOCX)

**S2 Table. Genes and oligonucleotides.** (DOCX)

## Acknowledgments

We gratefully acknowledge Dr. Ted Hackstadt (NIH/NIAID) for generously providing the modified targetron vector pARR and the expression plasmid pRAMF2 for our rickettsial protein expression studies. We are grateful to Dr. Chen Xing for providing guidance on CRISPR knockout experiments and generously sharing the lentiCRISPRv2, psPAX2, and pMD2.G plasmids.

## Author contributions

**Conceptualization:** Kehan Xu.

**Formal analysis:** Maozhang He, Kehan Xu.

**Funding acquisition:** Yan Liu, Kehan Xu.

**Investigation:** Yan Liu, Weiting Zhou, Jiaying Zhao, Yu Xin, Xuan Ouyang, Yonghui Yu, Jun Jiao, Kehan Xu.

**Methodology:** Weiting Zhou, Jiaying Zhao, Qingyin Shi, Yu Xin, Xuan Ouyang, Yonghui Yu, Jun Jiao, Yajun Song, Kehan Xu.

**Project administration:** Kehan Xu.

**Resources:** Yan Liu, Yajun Song, Kehan Xu.

**Supervision:** Yan Liu, Yajun Song, Kehan Xu.

**Validation:** Weiting Zhou, Jiaying Zhao, Yajun Song, Kehan Xu.

**Visualization:** Kehan Xu.

**Writing – original draft:** Kehan Xu.

**Writing – review & editing:** Yan Liu, Maozhang He, Yajun Song, Kehan Xu.

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
