## [Decision Letter · Decision Letter 0]

23 Mar 2025

PPATHOGENS-D-25-00114

VapC toxins promote the pathogenesis of Rickettsia heilongjiangensis by cleaving essential RNAs from both Rickettsia and its host

PLOS Pathogens

Dear Dr. Xu,

Thank you for submitting your manuscript to PLOS Pathogens. After careful consideration, we feel that it has merit but does not fully meet PLOS Pathogens's publication criteria as it currently stands. Therefore, we invite you to submit a revised version of the manuscript that addresses the points raised during the review process.

Please submit your revised manuscript within 60 days May 22 2025 11:59PM. If you will need more time than this to complete your revisions, please reply to this message or contact the journal office at plospathogens@plos.org. Please include the following items when submitting your revised manuscript:

We look forward to receiving your revised manuscript.

Kind regards,

Zhao-Qing Luo

Academic Editor

PLOS Pathogens

Matthew Wolfgang

Section Editor

PLOS Pathogens

 Sumita Bhaduri-McIntosh

Editor-in-Chief

PLOS Pathogens

orcid.org/0000-0003-2946-9497

 Michael Malim

Editor-in-Chief

PLOS Pathogens

orcid.org/0000-0002-7699-2064

**Journal Requirements:**

At this stage, the following Authors/Authors require contributions: Yan Liu, Weiting Zhou, Qingyin Shi, Yu Xin, Xuan Ouyang, Yonghui Yu, Jun Jiao, Yajun Song, and Kehan Xu. Please ensure that the full contributions of each author are acknowledged in the "Add/Edit/Remove Authors" section of our submission form.

- ® on page: 23

- TM on page: 23.

5) We notice that your supplementary Figures, and Tables are included in the manuscript file. Please remove them and upload them with the file type 'Supporting Information'. Please ensure that each Supporting Information file has a legend listed in the manuscript after the references list.

Potential Copyright Issues:

i) Please confirm (a) that you are the photographer of S4A, or (b) provide written permission from the photographer to publish the photo(s) under our CC BY 4.0 license.

ii) Figure 7I. Please confirm whether you drew the images / clip-art within the figure panels by hand. If you did not draw the images, please provide (a) a link to the source of the images or icons and their license / terms of use; or (b) written permission from the copyright holder to publish the images or icons under our CC BY 4.0 license. Alternatively, you may replace the images with open source alternatives. See these open source resources you may use to replace images / clip-art:

7) Please amend your detailed Financial Disclosure statement. This is published with the article. It must therefore be completed in full sentences and contain the exact wording you wish to be published.

2) If any authors received a salary from any of your funders, please state which authors and which funders..

**Reviewers' Comments:**

Reviewer's Responses to Questions

**Part I - Summary**

Reviewer #1: In this manuscript, Liu et al. characterize a newly discovered toxin-antitoxin system in Rickettsia that enables the bacteria to withstand host reactive oxygen species. The study elegantly demonstrates that these systems are expressed during the early stages of Rickettsia infection and that VapBC1 is secreted and plays a crucial role in pathogenesis and resistance to host ROS. The authors provide compelling evidence that VapBC1 exerts its function by cleaving the anti-codon loop of tRNAfMet, whereas VapBC2 nonspecifically degrades host rRNA. Overall, this is a well-executed and rigorously controlled study. My comments are minor and largely cosmetic.

Reviewer #2: Toxin-antitoxin (TA) modules enable bacteria to persist under stressful environments. In this manuscript, authors identified and characterized 2 TA system: vapBC1 and vapBC2 from an intracellular pathogen R. rickettsii. The data showed that vapBC1 is crucial for Rickettsia to withstand accumulated host reactive oxidative species (ROS), via induction of bacterial dormancy through cleavage on the anti-codon loop of tRNAfMet, thereby facilitating

intracellular survival and infection in a mouse model. Another vapBC module (vapBC2) was found to be activated and toxin exposed to host cytoplasm, contributing to Rickettsia's virulence and adaptability in its human host by non-specifically degrading host rRNAs rather than regulating rickettsial growth.

Overall, the experiments are well designed, data is solid, the discovery is novel and significant.

Minor comments:

1)In the result section "Rh-B8 VapBCs have distinctive intermolecular...". These data were generated from predicted models and interactions, the conclusion should be interpreted with caution (i.e., softening the conclusions)

2) What is homology level between vapB1 and vapB2?

3) in some result sections, VapC1 or VapC2 should be clearly indicated, rather than VapC toxins.

Reviewer #3: The manuscript by Liu et al. describes the identification and initial characterization of two toxin-anti-toxin modules, VapBC1 and VapBC2 in the spotted fever group Rickettsiae. Both modules were induced during infection with a matched induction of expression of the rickettsial Lon protease. The authors identified Lon as the protease responsible for cleaving the antitoxin modules VapB1 and VapB2, thus respectively freeing VapC1 and VapC2. The authors also provided evidence for the specific interactions of VapB1 with VapC1 and of VapB2 with VapC2. Both protease and VapBC modules were transcriptionally induced in response to exposure to H2O2. Additional characterization revealed the secretion of VapC2 to the host cell cytosol, where it cleaves a ribosomal RNA component, while VapC1 was retained in the bacterial where it apparently cleaves fMet-tRNA. The former was speculated to be involved in host cell cytotoxicity, while the latter was hypothesized to arrest bacterial growth in times of oxidative stress. Significance of the VapBC modules were characterized using two assays – plaque formation and quantification of plaque size and in a mouse model of infection where body weight, body temperature, and death were monitored.

Overall, the authors provided evidence that demonstrated the potential pathogenic significance of both VapBC1 and VapBC2, along with mechanisms of how the toxin components are liberated, i.e. through the proteolytic action of Lon on the respective anti-toxin components. With regards to the mechanism of VapC1 cleavage of fMet-tRNA, evidence for this mode of action was primarily obtained from tRNA profiling of E. coli ectopically expressing VapC1, which was followed by in vitro cleavage experiments focused on rickettsial fMet-tRNA, and the identification of cleavage products by northern blot using an fMet-tRNA-specific probe. This is crucial data, which I commend the authors for providing. The authors further strengthened the tRNA cleavage function of VapC1 by demonstrating the lack of growth effects of a predicted catalytically dead VapC1 in both E. coli and Rickettsia.

What the manuscript failed to establish convincingly is the possible cause-and-effect relationship between rickettsial growth inhibition mediated by VapC1-dependent fMet-tRNA degradation and reduced ROS burden. The authors attributed this reduction to host response, and not considering a reduction due to growth arrest of rickettsiae.

**Part II – Major Issues: Key Experiments Required for Acceptance**

Reviewer #1: None noted

Reviewer #2: Overall, the experiments are well designed, data is solid, the discovery is novel and significant.

Reviewer #3: Issues mentioned below are related to the need to clarify any potential link between bacterial growth arrest and reduced ROS burden. The authors attributed this to the induction of host response. The reduction could also be related to growth-arrested bacteria.

The experiments conducted to generate data in Fig. 3E, where the authors measured ROS burden in mock- and infected cells over time should be confirmed with their various mutant strains to clarify potential direct role of the pathogen in the reduction in ROS levels. ROS levels could be monitored early in infection (0-12 h) with respect to infection with wild type and various VapBC mutant strains. Currently, it’s not clear what causes this reduction in ROS burden. Is it due to reduced pathogen replication via VapC1 action, or is it related to the general inhibition of host translation caused by VapC2? Is there a correlation between the rates of rRNA cleavage and ROS burden?

Another crucial aspect of the model proposed that was not addressed was how the pathogen recovers, i.e. how normal growth rate is restored or how it exits from growth arrest/inhibition after oxidative stress attenuation by the VapBC system; and when recovery happens during infection. Replenishing of the fMet-tRNA pool should also be monitored in experiments that address recovery.

To further implicate an oxidative stress-related function for VapC1, mutant cells unable to generate reactive oxygen species in response to rickettsia infection could be used to evaluate the growth dynamics of wild type and mutant rickettsial strains.

With regards to the observed cytotoxicity of VapC2 on host cells, and subsequent data that demonstrated a role for VapC1 in the degradation of host rRNAs, it seems contradictory when viewed in the context of the proposed host response to minimize oxidative damage. The preservation of this response was based on the observed basal level of ROS burden at later time points in infection, and the expression of several markers of host proteins that deal with oxidative stress. Could this be due to growth arrest of rickettsia in the presence of oxidative stress? In other words, would a VapC1 mutant strain, which is predicted to be incapable of growth arrest sustain relatively high ROS burden levels?

**Part III – Minor Issues: Editorial and Data Presentation Modifications**

Reviewer #1: Figure 2A: The labeling needs improvement. What does "S# →" represent (β-sheet)? What does the "L" with the dotted bracket indicate? Additionally, in panels D and E, the VapC1 bands are difficult to discern. Would it be possible to add a quantification graph?

Figure S3: The gel contains multiple wells labeled 1–7, but their meaning is unclear. These labels should be explicitly defined either in the figure or the legend.

Figure 4G and H: These panels are very busy. Is there a more effective way to present this data for improved clarity?

Reviewer #2: see above

Reviewer #3: The manuscript would benefit from a more thorough review of grammar. There were several instances in the article where "the" should be omitted. Some examples are line 340 - "may promote the host cell damage...", line 412 - "with the prior analysis...",

Line 231, "extend" should be "extent".

Also, "promotor" should be corrected to "promoter".

PLOS authors have the option to publish the peer review history of their article (what does this mean? ). If published, this will include your full peer review and any attached files.

**Do you want your identity to be public for this peer review?** For information about this choice, including consent withdrawal, please see our Privacy Policy .

Reviewer #1: No

Reviewer #2: **Yes: ** xingmin Sun

Reviewer #3: No

**Figure resubmission:**
---

## [Decision Letter · Decision Letter 1]

15 Jul 2025

Dear Dr. Xu,

We are pleased to inform you that your manuscript 'VapC toxins promote the pathogenesis of Rickettsia heilongjiangensis by cleaving essential RNAs from both Rickettsia and its host' has been provisionally accepted for publication in PLOS Pathogens.

Best regards,

Zhao-Qing Luo

Academic Editor

PLOS Pathogens

Matthew Wolfgang

Section Editor

PLOS Pathogens

Sumita Bhaduri-McIntosh

Editor-in-Chief

PLOS Pathogens

orcid.org/0000-0003-2946-9497

Michael Malim

Editor-in-Chief

PLOS Pathogens

orcid.org/0000-0002-7699-2064

Reviewer Comments (if any, and for reference):

Reviewer's Responses to Questions

**Part I - Summary**

Reviewer #1: (No Response)

Reviewer #2: Revision addressed my comments.

Reviewer #3: The issues I raised in the first round of review were related to the need to strengthen the cause-effect relationship between the toxin-antitoxin system and how it is involved in attenuating the host cell response to infection, while simultaneously regulating pathogen growth in times of stress, namely oxidative stress. The study relied on demonstrating correlation for some important points. In this version, the authors addressed the issues. Overall, this is an exciting study that establishes a modulatory role at the host and pathogen fronts of the toxin-antitoxin system.

**Part II – Major Issues: Key Experiments Required for Acceptance**

Reviewer #1: (No Response)

Reviewer #2: none

Reviewer #3: (No Response)

**Part III – Minor Issues: Editorial and Data Presentation Modifications**

Reviewer #1: (No Response)

Reviewer #2: none

Reviewer #3: (No Response)

PLOS authors have the option to publish the peer review history of their article (what does this mean? ). If published, this will include your full peer review and any attached files.

**Do you want your identity to be public for this peer review?** For information about this choice, including consent withdrawal, please see our Privacy Policy .

Reviewer #1: No

Reviewer #2: No

Reviewer #3: No